# LamaH-CE | *La*rge-Sa*m*ple D*a*ta for *H*ydrology and Environmental Sciences for Central Europe

Christoph Klingler, Karsten Schulz, Mathew Herrnegger

Institute for Hydrology and Water Management, University of Natural Resources and Life Sciences, Vienna, 1190, Austria

*Correspondence to*: Christoph Klingler (christoph.klingler@boku.ac.at)

**Abstract.** Very large and comprehensive datasets are increasingly used in the field of hydrology. Large-sample studies provide insights into the hydrological cycle that might not be available with small-scale studies. LamaH-CE (*La*rge-Sa*m*ple D*a*ta for *H*ydrology) is a new dataset for large-sample studies and comparative hydrology in Central Europe. It covers the entire upper Danube to the state border Austria / Slovakia, as well as all other Austrian catchments including their foreign upstream areas.

LamaH covers an area of about 170 000 km² in 9 countries, ranging from lowland regions characterized by a continental climate to high alpine zones dominated by snow and ice. Consequently, a wide diversity of properties is present in the individual catchments. We represent this variability in 859 gauged catchments with over 60 catchment attributes, covering topography, climatology, hydrology, land cover, vegetation, soil and geological properties. LamaH further contains a collection of runoff time series as well as meteorological time series. These time series are provided with daily and hourly resolution. All

meteorological and the majority of runoff time series cover a span of over 35 years, which enables long-term analyses with a high temporal resolution. The runoff time series are classified by over 20 attributes including information about human impacts and indicators for data quality and completeness. The structure of LamaH is based on the well-known CAMELS datasets. In contrast, however, LamaH does not only consider independent basins, covering the full upstream area. Intermediate catchments are covered as well, which allows together with novel attributes to consider the hydrological network and river topology in

applications. We describe not only the used basic datasets and methodology of data preparation, but also focus on possible limitations and uncertainties. LamaH contains additionally results of a conceptual hydrological baseline model for checking plausibility of the inputs as well as benchmarking. Potential applications of LamaH are outlined as well, since it is intended to serve as a uniform data basis for further research. LamaH is available at https://doi.org/10.5281/zenodo.4525244 (Klingler et al., 2021).

**1 Introduction**

Hydrology and hydrological processes are characterized by high spatiotemporal variability. Runoff generation in small-scale, alpine catchments with steep and complex topography is dominated by different processes compared to lowland rivers with flat topography. The water balance in an energy-limited, humid catchment in Europe is completely different than, for example, in a water-limited catchment in dry (semi-) arid regions in Africa or Australia. A water droplet flowing via the Russian Lena

river into the Arctic Sea has a completely different biography than a water droplet from Rwanda in Central Africa, which reaches the Mediterranean Sea via the Nile after more than 6 600 km. Boundary conditions and major drivers for the differences are the catchment attributes, which can be described by characteristics regarding topography, hydro-climate, land cover, geology and soil conditions.

In order to deepen our understanding of the hydrological process and further increase the reliability of (hydrological) models,

it is necessary to account for this spatiotemporal variability in our approaches. A number of international initiatives (e.g. Distributed Model Intercomparison Project (DMIP; Smith et al., 2004); Inter-Sectoral Impact Model Intercomparison Project (ISI-MIP; Warszawski et al., 2014); Model Parameter Estimation Project (MOPEX; Duan et al., 2006) or Hydrological Ensemble Prediction Experiment (HEPEX; Schaake et al., 2007)) have been launched in recent decades with the aim to advance the prediction of hydrologic variables through comprehensive model benchmarking in different regions of the world.

New efforts strive for creating homogeneous and consistent datasets, which serve as a solid basis towards the development of new modelling approaches.

In this context, a trend towards more complete and extensive datasets is apparent: 1) Remote sensing has enabled consistent and global mapping of Earth's atmosphere and surface. 2) New software platforms or applications for obtaining and processing these mostly very data-intense (e.g. regarding data volumes) remote sensing products facilitate their applicability. Examples

of such platforms are "Google Earth Engine" (GEEa, 2021; GEEb, 2021; Gorelik et al., 2017; Klingler et al., 2020), the "Copernicus Open Access Hub" (COPa, 2021) or the "Copernicus Climate Data Store" (COPb, 2021). 3) There is growing awareness that our understanding of the complex hydrological processes can be deepened through "large-sample" studies (Gupta et al., 2014). Large-sample hydrology (LSH) includes information from a broad range of different watersheds in order to derive robust conclusions (Addor et al., 2019). Several research groups in different areas of hydrology have already focused

on LSH for this reason (e.g. Berghuijs et al., 2014; Blöschl et al., 2019a; Döll et al., 2016; Gudmundsson et al., 2019; Luke et al., 2017; Kuentz et al., 2017; Singh et al., 2014; Van Lanen et al., 2013). 4) Finally, data-driven models and deep learning approaches have recently gained significant attention in hydrology (Sit et al., 2020). Independent from the fact that these developments are controversially discussed (Nearing et al., 2020), their excellent performance in time series prediction, including in an ungauged setting (e.g. Kratzert et al., 2019a), is related to the ability of machine learning to identify patterns

and relationships in data (Kratzert et al., 2019b). These approaches however strongly depend on the availability of large-sample datasets (e.g. Kratzert et al., 2019a; 2019b; Kratzert et al., 2018).

Given the workload and scope of large-sample studies, it is reasonable to differentiate between dataset preparation and the subsequent investigation (i.e. publish the findings separately), which allows a more detailed description of the dataset and enables easier access. A selection of previously published "large-sample" datasets can be found in Table 1 in Gupta et al.

(2014). Other datasets for large-sample hydrologic applications include the Global Runoff Reconstruction (Ghiggi et al., 2019), Global Streamflow Indices and Metadata Archive (Do et al., 2018; Gudmundsson et al., 2018), HydroATLAS (Linke et al., 2019), HydroSHEDS (Lehner et al., 2008), and the CAMELS (CAtchment Attributes and MEteorology for Large-sample Studies; Addor et at., 2017a; and references in the following section) collection. The CAMELS datasets are characterized by

a consistent data preparation and consistent structure. Furthermore, potential limitations as well as uncertainties are discussed

there in detail. However, CAMELS only include data for independent catchments, covering the full upstream area, and not for an interconnected river network (Addor et al., 2019). The first CAMELS dataset was published by Addor et al. (2017a) and Newman et al. (2015) for the contiguous territory of the United States, containing data for 671 watersheds. Further CAMELS datasets for Chile (Alvarez-Garreton et al., 2018; 516 catchments), Brazil (Chagas et al., 2020; 897 catchments), and Great Britain (Coxon et al., 2020; 671 catchments) followed later. CAMELS datasets always represent a composite of

hydrometeorological time series and static catchment attributes aggregated to polygons, which cover the full upstream area. The question, how reasonable and applicable meteorological and catchment attributes are, when aggregated to the full upstream area, is critical, especially for large basins.

LamaH (*La*rge-*Sa*mple *Da*ta for *H*ydrology) is a new dataset for LSH (859 gauged catchments) in Central Europe and is generally based on the structure of the CAMELS datasets. LamaH therefore includes runoff time series, meteorological

forcings as well as static catchment attributes, but offers a few novelties. For example, LamaH includes a basin delineation that represents the inter-catchment area (difference area or intermediate catchments) of neighboring gauges, in addition to the usual basin delineation used in CAMELS datasets, which is equivalent to the topographic (delineated only considering terrain features and ignoring potential subsurface cross-basin flows) catchment area of the individual gauges. Supplementary attributes such as the gauge topology, as well as the flow length and gradient between two adjacent gauges, are added to specify the

interconnected hydrologic network. This enables, for example, to model the local runoff generation in the intermediate catchments and the river routing separately. A further novelty of LamaH is the finer resolution of the provided hydrometeorological time series (daily and hourly). Time series with hourly resolution are crucial for a reliable result when modelling, for instance, the river routing or snow or glacier driven processes, where the observed signal in runoff shows a distinct diurnal pattern.

This paper is organized as follows: After a description of the project area (section 2) and included basins and catchments (section 3), the preparation of the hydrometeorological time series are described in section 4. Section 5 is about static catchment attributes and shows their spatial distribution. The set-up as well as the results of a hydrological baseline model are described in section 6. Additionally, uncertainties, limitations and restrictions of the used data sources or model outputs are discussed. Finally, section 7 includes a summary and an outlook of possible applications of LamaH.

**2 Domain of coverage**

LamaH covers an area of about 170 000 km² in 9 countries in Central Europe (Austria, Germany, Czech Republic, Switzerland, Slovakia, Italy, Liechtenstein, Slovenia and Hungary; sorted by descending contributing area). Its scope includes the upper Danube to the Austrian / Slovakian border, as well as all other catchment areas in Austria, including their adjacent upstream areas in neighboring countries. The Piz Bernina with 4 049 m a.s.l. represents the highest point within the project area, while

the lowest point at about 130 m a.s.l. is located at the most downstream gauge of the Austrian Danube. The dominant river is

the Danube (ICPDR, 2020; Prohaska et al., 2020), which has its source in the far west of the project area near Donaueschingen (Fig. 1, 8.2°E / 48.1°N). The catchments of Danube's main tributaries serve to divide the project area into 18 river regions (Table B1). An overview of the domain covered in LamaH with the river regions and the runoff gauges with their elevations is illustrated in Fig. 1. All river regions in the project except regions 1 and 11 are part of the Danube´s catchment. Water from

regions declared with "Danube B" in Fig. 1 joins the Danube outside the project area in Hungary or Croatia. River region 1 covers the upper catchment of the Rhine from its sources to Lake Constance ("Rhine") and region 11 covers the Austrian catchment area of the Vltava, which is the largest tributary of the Elbe ("Elbe").

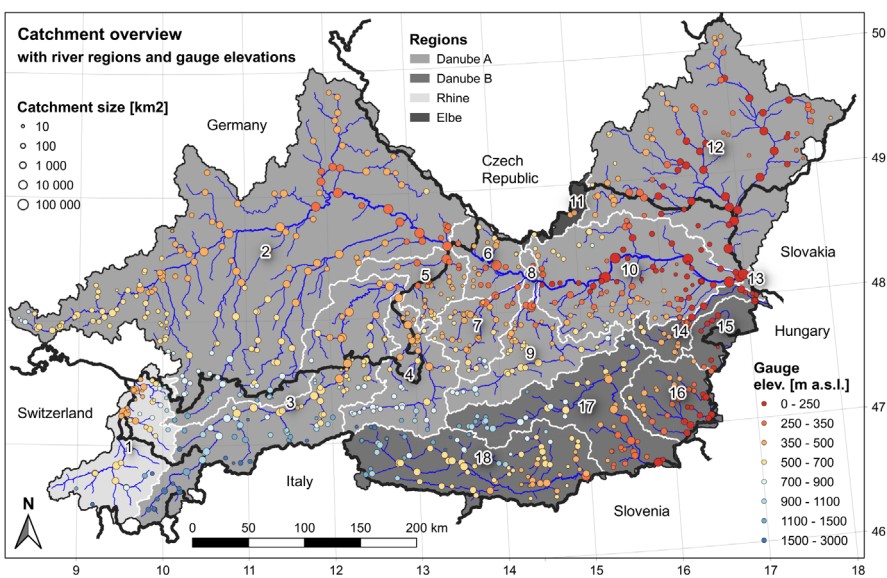

**Fig. 1: Overview of the area covered in LamaH (grey tones), and the runoff gauges with gauge elevation (circle color) and catchment**
**area (circle size). LamaH is divided into different river regions, which are bordered by the white lines. The black numbers are abbreviations for the individual regions, which are indicated in Table B1. The national borders are shown as thick black lines. Source of stream network: HydroATLAS (Linke et al., 2019). © EuroGeographics for the administrative boundaries.**

## 3 Basin delineations and aggregation approaches

Most meteorological time series and catchment attributes included in LamaH are based on global datasets, which were provided
either in raster or vector form. In LamaH, a catchment property or time step of a meteorological time series usually represents the mean computed across the topographic catchment of a gauge. Starting point for creating the aggregation polygons (catchments) were sub-basins from the datasets Digital Hydrological Atlas of Austria (HAO, 2007; full expansion) and HydroATLAS (Linke et al., 2019; level 12), which was used for areas not covered by HAO. The sub-basin outlets of HAO agree with the gauge locations. In contrast, the catchment boundaries of HydroATLAS were partially manually adjusted to
guarantee that the basin outlets of the polygons agree with the gauging station locations. Since the sub-basin delineation in HAO and HydroATLAS were aggregated to represent the complete topographic catchment area upstream of a gauge, the different resolutions in the data sets did not matter. We refer to this method of basin delineation in the further text and the

dataset as "basin delineation A" (Fig. 2a). Plausibility of this type of basin delineation was checked by calculating the ratio between the area of the aggregated basins and the officially, e.g. in the metadata of the gauges, declared catchment area ("area_ratio" in Table A1). The range of "area_ratio" lies between 0.89 and 1.34, with a standard deviation of 0.026. Catchments with larger deviation in area were manually checked and corrected if there was an obvious error. The median basin size over all 859 catchments of LamaH applying basin delineation A is 178 km², with a range of 4 to 131 247 km². Basin delineation A is identical to the delineation used in the CAMELS datasets. The advantage of basin delineation A is the independency between the basins, since the aggregation area fully represents the topographic catchment area of a gauge. However, for gauges with larger catchments, aggregation with basin delineation A leads to a significant loss of information, as variability as well as small-scale characteristics are lost.

Therefore, basin delineation A is supplemented by a form of delineation (basin delineation B, 859 catchments) where the topographic catchment area of the next upstream gauge (may be none, one, or more) is subtracted from that of the current gauge (Fig. 2b). This results in the representation of intermediate catchments, which become part of a large connected river network. The dependency among these intermediate catchments requires a catchment or gauge hierarchy (Fig. 2b, "HIERARCHY" in Table A1), as well as information regarding the upstream-downstream relationship ("NEXTUPID" or "NEXTDOWNID" in Table A1). The median basin size resulting from basin delineation B is 114 km², with a range of 1 to 2 500 km². Significant reduction of polygon size at the upper end ensures a more representative mapping of local features.

The third basin delineation provided in LamaH (further referred to as basin delineation C in the text and dataset) is similar to basin delineation B, but only includes catchments with no or only low anthropogenic influence (454 catchments; Fig. 2c). This provides a bundled collection of catchments that exhibit hydrological conditions that are close to natural ones. Anthropogenic influences in the catchments and runoff data is described more detailed in section 5.8.

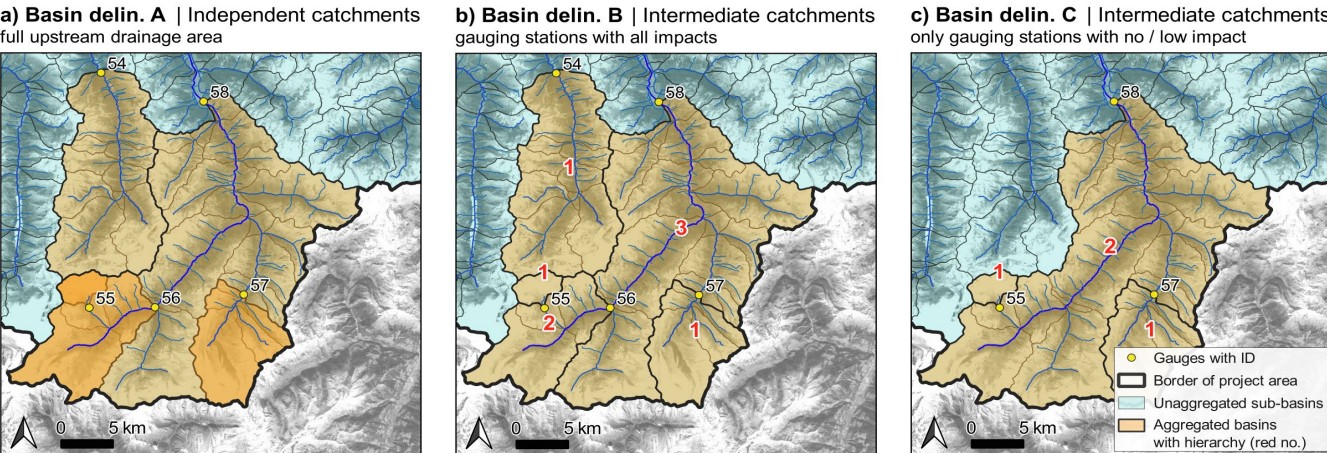

**Fig. 2: Types of basin delineations in LamaH shown with an example. Plot a) Basin delineation A (similar to the well-known CAMELS datasets): The aggregation area corresponds to the topographic catchment area of a gauge. In plot a), the aggregation area of gauges 56 and 57 overlaps with that of gauge 58, and the aggregation area of gauge 55 overlaps with that of gauges 56 and 58 (indicated by the different color tones). Plot b) Basin delineation B: The aggregation areas in this method considers the difference**

**area (intermediate catchments) between the topographic catchment area of the respective gauge and the catchment area of the next upstream gauges. Consequently, there are no overlaps, but a gauge hierarchy is necessary. The hierarchy of the gauges 54, 55 and 57 is 1, because there is no upstream gauge. Gauge 56 has the hierarchy 2, because gauge 55 with hierarchy 1 is upstream. Hierarchy 3 is assigned to gauge 58, because there is at least one gauge with hierarchy 2 (gauge 56) in the upstream area. Plot c) Basin delineation C: Similar to basin delineation B, but only uninfluenced or low-influenced gauges / catchments (section 5.8) are considered. In plot c), it is assumed that gauges 54 and 56 are strongly influenced. Consequently, these two gauges are excluded from the basin delineation. The aggregation area of gauge 58 (now hierarchy 2) includes the intermediate catchment area of gauge 56. Source of background satellite image: Google © 2020 TerraMetrics, Kartendaten © 2020. Source of stream network: TYROL (2020).**

Aggregation of the spatially distributed information of the used basic datasets for meteorological time series and various static attributes is performed for each of the 3 basin delineations by calculating the area-weighted arithmetic mean (otherwise indicated in the text). This method of aggregation is used for coarser gridded and vectoral data sources and is referred in the following as "upscaling approach 1". The alternative "upscaling approach 2" is based on all the raster cells, which centroids are located inside the polygon ("aggregated basins" in Fig. 2). In case of small catchments, where no or only one raster centroid intersects the polygon, upscaling approach 1 was used. Upscaling approach 2 is mainly used for relatively fine-gridded data sources (< 1 km grid size), since it is not that computing-intensive and potential inaccuracies are negligible. The applied approach is indicated in the relevant tables in appendix A.

## 4 Hydrometeorological time series

### 4.1 Runoff data

LamaH contains daily and hourly runoff time series for 882 gauges, located in 4 countries (Austria, Germany, Switzerland and Czech Republic). The difference to the 859 catchments defined in basin delineation A (section 3) can be explained by the fact that 23 gauges, which mostly do not have a clearly definable catchment area (e.g. gauges at artificial channels or below large karst springs; section 5.8), were not considered in basin delineation. The main provider of runoff time series was the Hydrographic Central Bureau of Austria (HZB, 2020), which contributed data for 609 gauges located in Austria. The hydrographical services of the German federal states Bavaria (GKD, 2020) and Baden-Württemberg (LUBW, 2020) provided 125 and 61 runoff time series, respectively. 25 runoff time series came from the hydrological office of Switzerland (BAFU, 2020), while time series for 61 gauges were provided by the Czech Hydrometeorological Institute (CHMI, 2020). The format of all obtained time series was unified, enabling much easier data processing. The various gauge attributes and metadata are listed and described in Table A1. The unit of discharge is $m^3 \, s^{-1}$ for both daily and hourly resolution. Conversion to runoff heights can be performed using the catchment area provided ("area_gov" in Table A1 or "area_calc" in Table A3).

Runoff time series are in most cases derived by water level - discharge relationships (rating curves). Changes in channel profile, e.g. after floods with strong bedload transport, extrapolation of the rating curve or backwater effects and transient runoff conditions (runoff hysteresis) can lead to an incorrect runoff determination (McMillan et al., 2012). However, attempts are usually made to minimize this source of error by periodically adjusting the rating curve. The adjustment frequency of these

rating curves is not publicly available, but only gauges in the highest quality class (quality classes are declared in Bavaria and the Czech Republic) were included in LamaH.

Runoff time series with daily resolution are often provided with longer observation periods than those with hourly resolution. Therefore, daily and hourly runoff time series can be obtained separately from the listed hydrological offices. However, we normally requested only the time series with hourly resolution and derived the daily time series from them. Thereby the hourly values of the respective day were used for determining the daily values (as well as for the meteorological variables), e.g. for 1 October 1981: 1 October 1981 00:00 to 1 October 1981 23:59 GMT. This approach was chosen for the runoff data from Austria, Germany and Switzerland, since those time series with hourly resolution mostly include quite long recording periods.

Fig. 3a and the included histogram show that most gauges have continuous data recording since the late 1970`s. In contrast, the time series from the Czech Republic were requested with both daily and hourly resolution, as the continuous (hourly) time series here only starts after 2005. The runoff time series in LamaH were limited to the period 1981 to 2017, because 1981 was the starting year of the meteorological ERA5-Land forcings (section 4.2), and 2017 was the last year for quality-controlled runoff data from Austria at the point of request.

Although the exact scope of data verification by the staff of the various hydrological services is not further specified, we have added an attribute describing the check status ("ckhs" in Table A1) to each time step of the runoff time series. The Austrian, Czech and Swiss runoff data are provided exclusively checked, while the runoff data from the Bavarian hydrographic service is in most cases quality controlled until the years 2014 / 2016. Data from the German federal state Baden-Württemberg is often checked only from the year 2010 onwards. Some time series included gaps, even after checking by the hydrological services ("gaps_pre" in Table A1). Gaps of up to 6 hours were filled with linear interpolation during our processing, if the number of consecutive gaps was less than 7. Any remaining gaps (> 6h) were marked with the number -999. The fraction of remaining gaps in the continuous runoff time series is declared by the attribute "gaps_post" and illustrated in Fig. 3b. It is shown that those gauges with very few gaps (< 0.1‰) are mostly located in Austria, Czech Republic and Switzerland. About 80% of the 882 gauges have no gaps in their continuous time series after our processing. The time steps with gaps before our processing are listed in separate files, attached to the dataset. The spatial distribution of the gauge hierarchy (see caption of Fig. 2b) is mapped in Fig. 3c., where 50% of all gauges have a hierarchy of 1. The highest hierarchy (26) is found for the very last downstream gauge of the Austrian Danube (ID 399). Lastly, the attributes "nrs_euhyd" and "nrs_rivat" allow cross-references to the river network datasets EU-Hydro-River Network Database (EEA, 2019) and RiverATLAS (Linke et al., 2019), respectively. Thereby the ID of the corresponding river section to the gauge is given, enabling access to their attributes and routing through these networks.

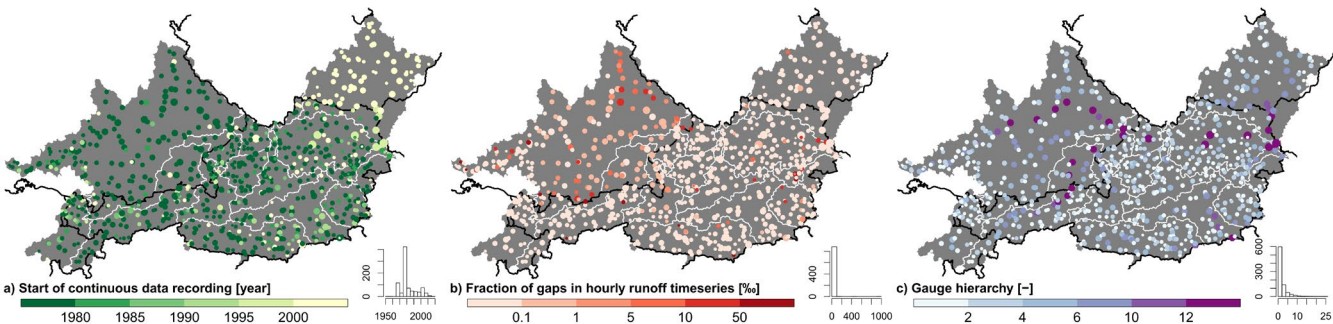

**Fig. 3: Maps showing a selection of gauge-referenced attributes. The size of the circles is proportional to the respective catchment area. The histograms indicate the number of gauges (out of 859) in each category. © EuroGeographics for the administrative boundaries.**

## 4.2 Meteorological data

Given the extent of the ECMWF (European Centre for Medium-Range Weather Forecasts) ERA5-Land dataset with global coverage (Muñoz Sabater et al., 2021), it was possible to obtain gap-free time series with daily and hourly resolution for 15 meteorological variables and 39 years (Table A2). ERA5-Land is a derivative of the ERA5 climate reanalysis (Hersbach et al., 2020), but only covers the terrestrial components. Further developments compared to ERA5 include an interpolation package for a finer temporal resolution, an additional sea level adjustment of the meteorological fields, as well as more efficient possibilities for the import of updates (Muñoz Sabater et al., 2021; Yang and Giusti, 2020). ERA5-Land has a spatial resolution of 0.1 arc degrees (about 9 x 11 km at the latitudes of the project area) compared to the grid size of ERA5 of 0.25 arc degrees. The temporal resolution of ERA5-Land is 1 hour, while ERA5 only has a 3-hour resolution. There is no data assimilation (fitting to observations) applied for ERA5-Land, but observations are indirectly implemented via the assimilated atmospheric fields of ERA5 (Hennermann and Guillory, 2020; Yang and Giusti, 2020). In accordance to ECMWF regulations, an uncertainty estimate for ERA5-Land will be released (Muñoz Sabater, 2019b; Muñoz Sabater et al., 2017), but was not available at the time of writing (January 2021).

Meteorological time series were computed for all 3 forms of basin delineation (A/B/C in section 3). The aggregation was done by calculating the area-weighted arithmetic mean (upscaling approach 1). As already mentioned in the introduction, we would like to point out possible uncertainties of the published data. We therefore determined the components of the water balance for the period 1 October 1989 to 30 September 2009 (hydrological years 1990 to 2009) and plotted them (Fig. 4a). Values of catchments influenced by cross-basin water transfers, water withdrawals or intakes, large karstic springs or high infiltration (section 5.8) are not shown in Fig. 4a/c to allow a more objective interpretation. In case of long-term water balances, it is usually feasible to neglect artificial storage in the catchment. The difference between long-term mean precipitation (P) and runoff height (Q), as recorded at the gauging station, should be equal to the total evapotranspiration (ETA) in a fulfilled water balance. This would be shown by having all points in Fig. 4a on the 1:1 line, which is not the case. Reasons for the rather strong scatter (Pearson correlation R = 0.30) may be  an insufficient representation of precipitation or total evapotranspiration by ERA5-Land, an inaccurate recording of runoff (e.g. strong, unrecorded groundwater flow or change in river profile at the

235 gauging station and thus inadequate water level - discharge relationship), a significant deviation between topographic and hydrographic catchment area (subsurface inflows and outflows, especially in karstic areas), or lastly, in case of existing glaciers, a negative mass balance (Lambrecht and Kuhn, 2007; Kuhn, 2004; Kobolschnig and Schöner, 2011; Oerlemans et al., 1998; WGMS, 2005). Using other precipitation datasets for the same evaluation does not result in a significantly more compliant long-term water balance. CHIRPS Daily v2 (Funk et al., 2015) resulted in a correlation R between (P - Q) and ETA

of 0.34 or MSWEP v2.2 (Beck et al., 2017; 2019) in even a lower R-value of 0.26. Even if we cannot resolve the issue at hand, the total evapotranspiration from ERA5-Land and its dependence on elevation seems quite plausible compared to other studies (Fig. 4a; HAO, 2007, Map 3.3; Herrnegger et al., 2012, Fig. 20). Negative differences of mean precipitation and runoff height (Fig. 4a) and thus runoff coefficients > 1.0 (Fig. 4c, 32 of 594 catchments) are mainly present in higher terrain (negative mass balance of glaciers, Fig. 8d) and in catchments with a high fraction of carbonate sedimentary rocks (indicator of karst, Fig.

11c). Since ERA5-Land indirectly incorporates in-situ observational data via the assimilated atmospheric fields of ERA5 (Yang and Giusti, 2020), systematic measuring error of a terrestrial station being used could explain insufficient mean precipitation (Herrnegger et al., 2018). The individual components of the water balance are attached to the dataset for every catchment (Table A10), since this evaluation might be useful for explaining any deviations in a later modelling.

The Budyko curve (Fig. 4b; Budyko 1974) describes the relationship between the ratio current evapotranspiration /

precipitation (ETA/P) and the ratio potential evapotranspiration / precipitation (PET/P) and indicates whether evapotranspiration of a catchment is limited by energy or water. Ideally all points should lie in the proximity of the Budyko curve. The deviation from this ideal case can primarily be explained by a significantly too high PET of ERA5-Land over nearly the entire range of elevation. For example, 98% of all 859 watersheds show mean annual PET sums above 1000 mm ("PET" in Table A10). As these PET sums are not realistic at the latitudes of the project area (HAO, 2007, Map 3.2; Herrnegger et al.,

2012, Fig. 17), we did not include PET time series of ERA5-Land in the LamaH dataset. However, there is the possibility to obtain daily PET time series from provided fluxes of the hydrological model (section 6; Table C2). The runoff coefficient (Q/P) as a function of the ratio mean precipitation / total evapotranspiration (P/ETA) is shown in Fig. 4c. The altitudinal dependency can be clearly seen in Fig. 4c, while catchments with lower mean elevation show less scatter. Fig. 4d shows the contrast of the long-term precipitation of ERA5-Land with those of ERA5 (Pearson correlation R = 0.936). ERA5 indicates

systematic surplus of precipitation at catchments with mean altitudes above 2000 m a.s.l. compared to ERA5-Land, while at catchments with mean altitudes between 800 and 1200 m a.s.l. the opposite is more likely to be the case. The correlation between the long-term precipitation of ERA5-Land and those of the dataset CHIRPS daily v2 (Funk et al., 2015) is 0.916 (Fig. 4e). Further, the mean precipitation sums of CHIRPS daily v2 tend to be lower than those of ERA5-Land over the whole range of altitudes. More scatter (R = 0.841) appears especially in catchments with higher elevations when comparing the long-term

precipitation sums of ERA5-Land and MSWEP v2.2 (Beck et al., 2017; 2019; Fig. 4f).

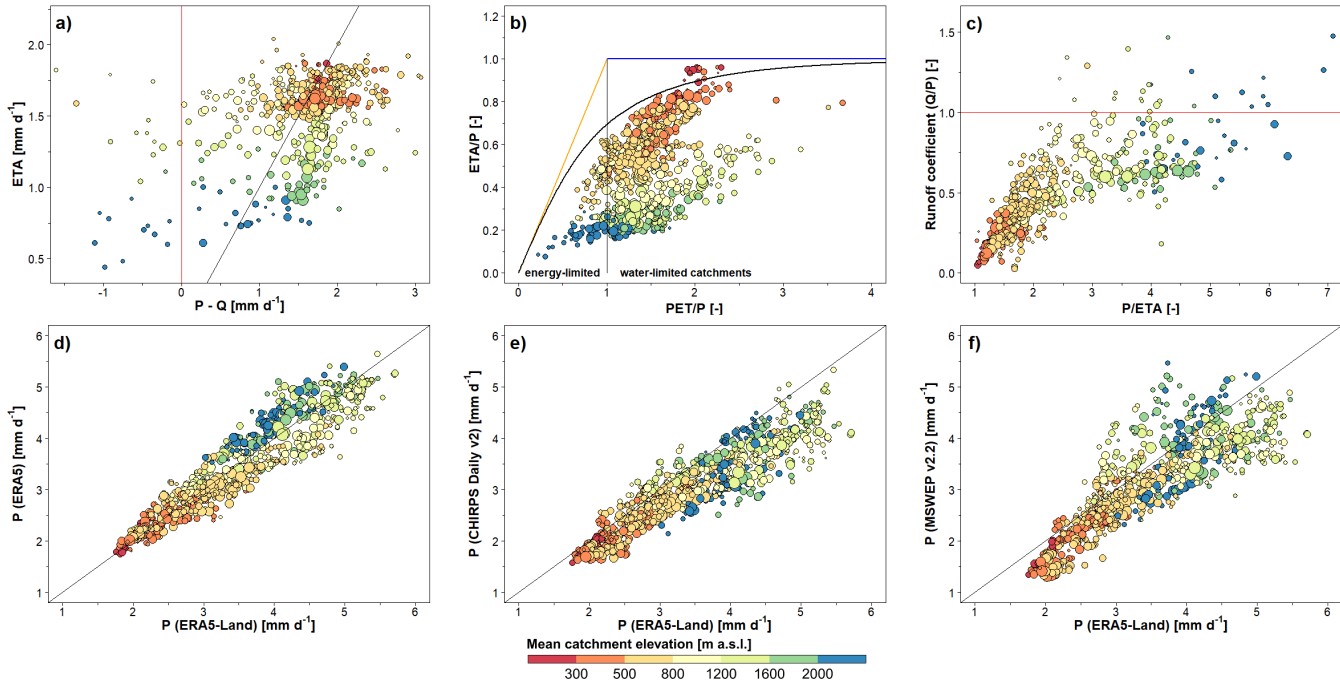

**Fig. 4: Analysis regarding the long-term water balance, evapotranspiration and comparison of the ERA5-Land`s mean precipitation with other datasets for the hydrological years 1990 - 2009 and basin delineation A. a) Total evapotranspiration (ETA) from ERA5-Land as a function of the difference between precipitation (P) from ERA5-Land and recorded runoff depth (Q). b) Budyko curve indicates if ET of a catchment is limited by energy (PET/P < 1) or by water (PET/P > 1). c) shows the runoff coefficient (ratio of Q and P) as a function of the fraction of ERA5-Land´s precipitation and total evapotranspiration. In a), b) and c), values are only plotted for basins with observations for the period 1 October 1989 to 30 September 2009 (717 basins). Further, in a) and c), values are only plotted for basins not affected by artificial water input or withdrawal, karstic springs or high infiltration (594 basins; section 5.8). Plots d) / e) and f) illustrates the relationship between ERA5-Land´s precipitation compared to the datasets "ERA5", "CHIRPS Daily v2.0" and "MSWEP v2.2" in 859 catchments. The diagonal black line in a), d), e) and f) is the 1:1 line. The red lines in a) and c) show physical constraints. The sloped orange line in b) indicates the energy-limit, while the horizontal blue line represents the water-limit. The curved black line in b) represents the Budyko curve. The size of the symbols in all plots is proportional to the catchment area, while the color indicates the mean elevation of the catchment (see legend at bottom).**

## 5 Catchment attributes

The various physio-geographical characteristics of a catchment, as well as their interactions, are essential for water storage and transport on and below earth's surface (Blöschl et al., 2013). The spectrum of influencing catchment characteristics includes topography, climate, hydrology, land cover, vegetation, soil, geology, as well as the type and degree of (anthropogenic) impact on runoff processes. Furthermore, catchment attributes are crucial to determine interrelationships among different watersheds along several gradients (Addor et al., 2017a; Falkenmark and Chapman, 1989; Fan et al., 2019).

In most cases, we used freely available datasets with global or at least European coverage for deriving the different catchment attributes. The used datasets for deriving the attributes, methods of processing, possible uncertainties as well as the spatial distribution of catchment attributes (Addor, 2017b) are discussed in more detail in the following subsections. It is clear that

due to the large dataset this account is far from complete. The individual attributes are listed in tabular form in the appendix with a more detailed description, units and with reference to the data sources.

## 5.1 Topographic indices

We calculated 10 topographic attributes, which are listened in Table A3. The attribute "area_calc" describes the calculated aggregation (catchment) area, depending on the applied method of basin delineation (section 3). A key factor for hydrologic processes is elevation, as it affects numerous other catchment characteristics including climate, land cover, vegetation, or soil development (Addor et al., 2017a). We derived the mean catchment elevation (Fig. 5b, "elev_mean" in Table A3), the median elevation ("elev_med"), standard deviation within a catchment ("elev_std"), the elevation range (maximum-minimum elevation in the catchment, Fig. 5c, "elev_ran"), as well as the mean catchment slope (Fig. 5d, "slope_mean") from NASA's SRTM dataset (Farr et al., 2007). SRTM features a grid size of 30 m and provides a maximum global absolute vertical error of 16 m at a 90% confidence interval, while accuracy decreases with increasing elevation and slope (Farr et al., 2007). The slope was calculated with the algorithm of Horn (1981) using the terrain elevation from SRTM. High mean catchment elevations and slopes are most apparent in the Eastern Alps, which extend from the southwest to the central east of the project area. This high elevated area is mainly surrounded by the flatter Alpine foothills and regions with older geological zones (Fig. 5b).

The shape of the catchment area and the stream network influence runoff formation. The direction of precipitation in relation to the longitudinal axis of the catchment is of major interest in case of flood situations, especially in larger catchments. For this reason, we specified the angle between the north direction and the longitudinal axis ("mvert_ang") in addition to the distance of the longitudinal axis of a catchment ("mvert_dist"). In combination with the two wind components of ERA5-Land ("10m_wind_u", "10m_wind_v" in Table A2) it is possible to derive the relative rainfall trajectory. Furthermore, the catchment shape is relevant for the rise of the flood wave. The attribute of length elongation according to Schumm (1956) (Fig. 5e, "elon_ratio") is an indicator regarding the "roundness" (the higher, the rounder) of the catchment. Stream density (Fig. 5f, "strm_dens") is a function of several characteristics (e.g. climate, relief, soil properties, geology, vegetation, land use, glaciation or karstification) and can therefore be an informative indicator for comparing watersheds (Olden and Poff, 2003). The EU-Hydro – River Network Database (EEA, 2019) is used for calculating the stream density, since it is a fine-resolved dataset and consistent over the covered project area.

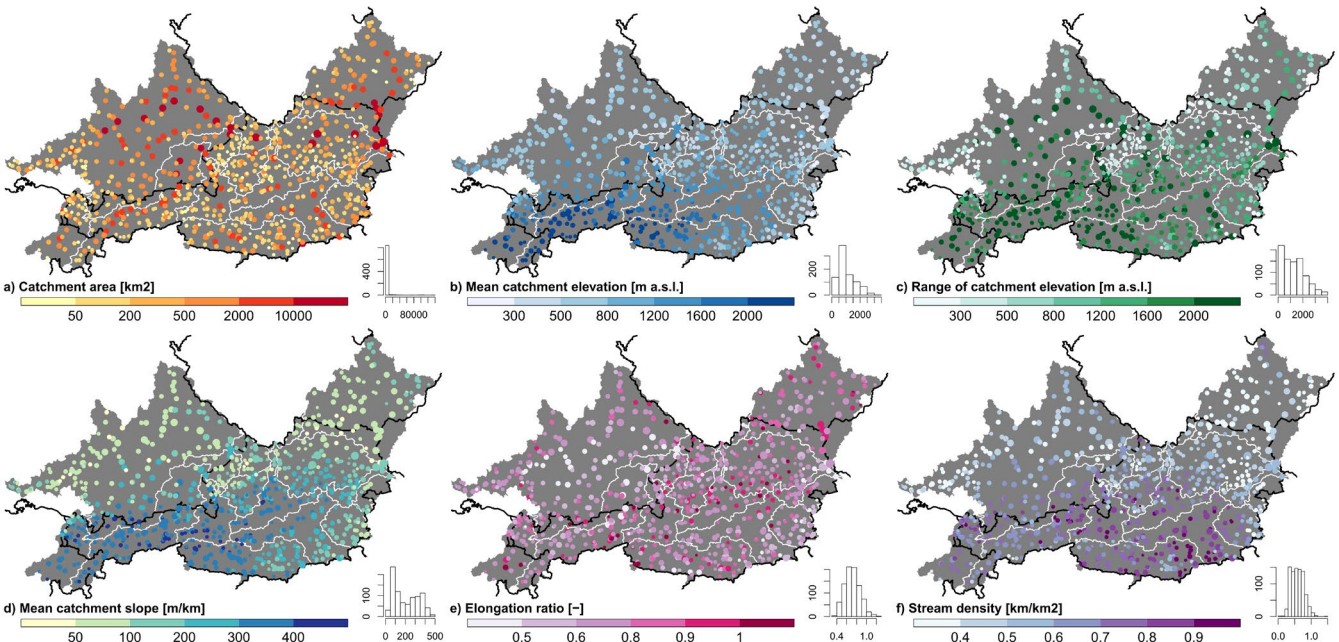

**Fig. 5: Spatial distribution of a selection of topographic attributes representing the characteristics of the entire topographic catchment (basin delineation A, Fig. 2a). The histograms indicate the number of basins (out of 859) in each category. The size of the circles is proportional to the catchment area. © EuroGeographics for the administrative boundaries.**

### 5.2 Climatic indices

LamaH includes 12 attributes reflecting aspects of climatic characteristics (Table A4). These attributes were calculated mainly from the meteorological time series of ERA5-Land for the period 1 October 1989 to 30 September 2009 (Addor, 2017b). The Reference Evapotranspiration (ET0) from the "Global Aridity Index and Potential Evapotranspiration (ET0) Climate Database v2" (GCD v2; Trabucco and Zomer, 2019), which was computed for the period 1970 to 2000 is provided as an alternative to ERA5-Land´s potential evapotranspiration, which shows unrealistic high values (section 4.2). ET0 describes the atmosphere's capacity for evapotranspiration given defined vegetation characteristics. Potential evapotranspiration (PET) can be derived from ET0 using correction factors for vegetation and soil properties (Allen et al., 1998; Hargreaves, 1994), but was not realized in LamaH.

Long-term climatic characteristics are described by long-term daily precipitation (Fig. 6a, "p_mean" in Table A4), reference evapotranspiration (Fig. 6b, "et0_mean"), total evapotranspiration ("eta_mean"), and the aridity index (Fig. 6c, "arid_2"). As an alternative, the aridity index which was calculated by dividing ET0 from GCD v2 and precipitation from ERA5-Land is also included ("arid_1"). The spatial pattern of long-term precipitation sums (Fig. 6a) clearly shows an elevation gradient and blocking effects along the northern Alps. The west of the project area is characterized by higher mean precipitation due to the stronger influence of oceanic climate. The relationship between mean catchment elevation (Fig. 5b) and ET0 (Fig. 6b, Pearson correlation R = -0.79), aridity (Fig. 6c, R = -0.68), or the fraction of precipitation falling as snow (Fig. 6g, R = 0.96) show

similar spatial patterns. About 19% of all catchments, which are exclusively located in the eastern part of the project area, have

aridity greater than 1 ("arid_2").

Attributes characterizing seasonality are the fraction of precipitation falling as snow (Fig. 6g, "frac_snow") and the seasonality index, which relies on sinusoids to describe the precipitation cycle over the year (Fig. 6d, "p_season"). A higher positive seasonality index indicates higher precipitation sums during summer, while values near 0 show a more balanced precipitation distribution throughout the year.

While long-term and seasonal indices describe general climatology, they provide little or no information about relatively short-term events such as drought or heavy rainfall. Consequently, we calculated attributes representing the frequency of high precipitation days (days per year with at least 5 times mean daily precipitation; Fig. 6e, "hi_prec_fr") and dry days (days per year with max. 1 mm d$^{-1}$ precipitation; Fig. 6h, "lo_prec_fr"), their mean duration (Fig. 6f, "hi_prec_du" / Fig. 6i, "lo_prec_du"), and the most likely season of occurrence ("hi_prec_ti" / "lo_prec_ti"). The reason for the higher frequency of

high precipitation days in the south-eastern part of the project area (Fig. 6e) is primarily the combination of relatively rainfall-rich convective precipitation events during the summer months and relatively low precipitation sums during the rest of the year (Fig. 6d). For both, the mean frequency of dry days (Fig. 6h, R = -0.62) as well as their mean duration (Fig. 6i, R = -0.57), a negative spatial correlation with the mean catchment elevation (Fig. 5b) can be observed. Most common season for high precipitation is for 89% of all 859 catchments summer (June, July and August), while winter (December, January and February)

is the most common season for dry days in 89% of the basins.

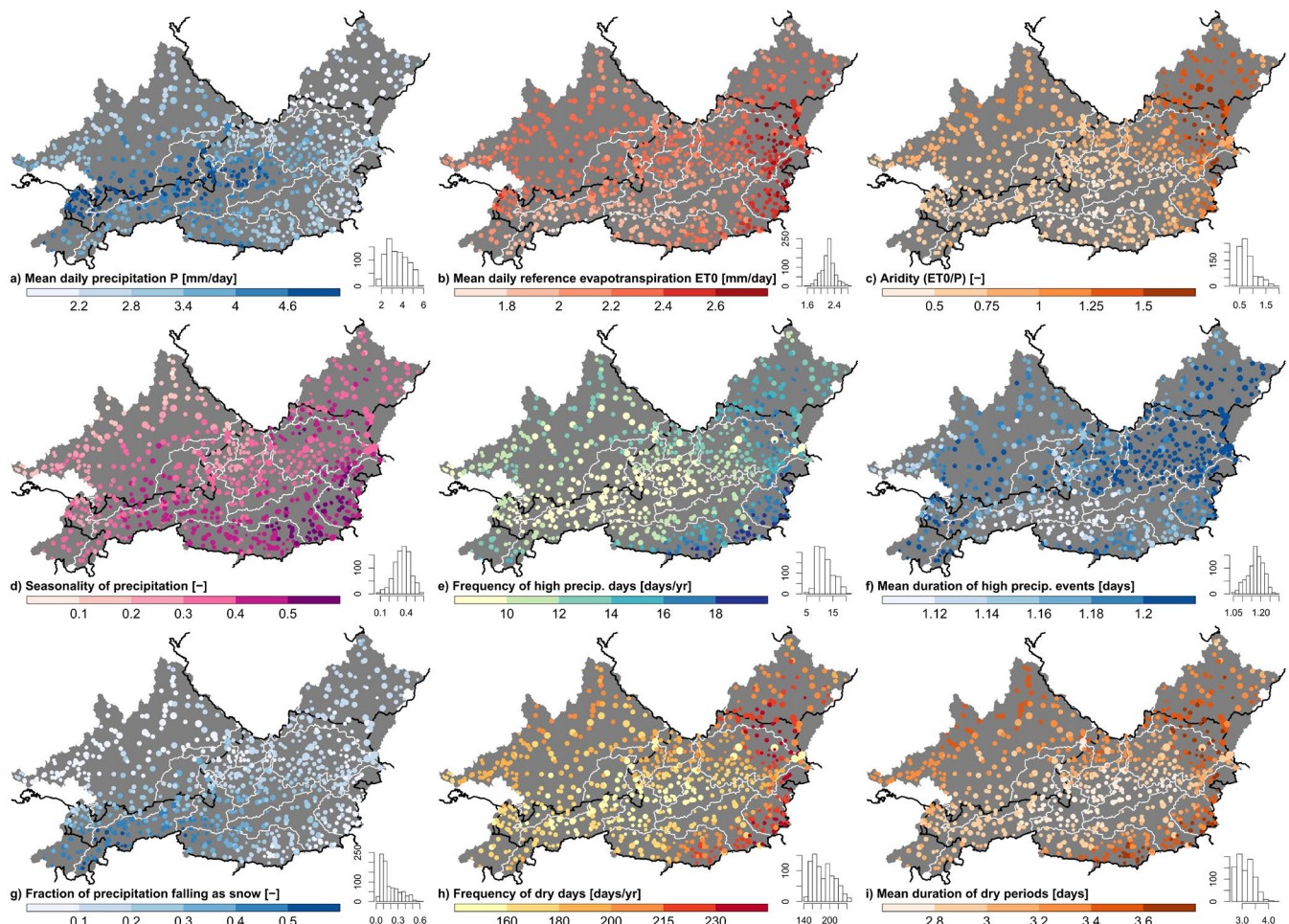

**Fig. 6: Spatial distribution of a selection of climate indices representing the characteristics of the entire topographic catchment (basin delineation A, Fig. 2a). The histograms indicate the number of basins (out of 859) in each category. The size of the circles is proportional to the catchment area. © EuroGeographics for the administrative boundaries.**

## 5.3 Hydrological signatures

The runoff time series are characterized by 14 attributes (Table A5) which were calculated for the period 1 October 1989 to 30 September 2009 (Addor, 2017b). The indices were computed for those gauges which cover the whole period of investigation (717 gauges). However, evaluations for the entire period of record (first 1 October after 1981 to 30 September 2017) are additionally made available if at least 5 full hydrological years are recorded. 4 gauges do not meet this requirement. Hydrological signatures are calculated, only if the fraction of gaps is less than 5% for both evaluation periods. Hydrological attributes can be divided into those describing long-term characteristics, seasonality, and more short-term situations such as high and low flow.

Aridity by itself can be a good predictor for runoff occurrence in a catchment (Arora, 2002; Blöschl et al., 2013; Budyko, 1974). This is shown by the similar spatial pattern of long-term runoff height (Fig. 7a, "q_mean" in Table A5, R = -0.61), and

runoff ratio (Fig. 7b, "runoff_ratio", R = -0.60) compared to those of aridity (Fig. 6c). The runoff coefficient (Q/P) is the fraction of precipitation that drains a surface after deducting evapotranspiration, groundwater flow or change in storage in the long-term. Explanations for runoff coefficients greater than 1 are given in section 4.2. The ratio of baseflow and total runoff can be a useful indicator for watershed classification (Sawicz et al., 2011; Fan, 2015), and is further referred as the baseflow index ("baseflow_index"). It should be noted that this index is highly dependent on the method used to separate the hydrograph (Beck et al., 2013; Chapman, 1999; Eckhardt, 2008). For this reason, we used the Ladson filter (Ladson et al., 2013) and the approach of Tallaksen and Van Lanen (2004) for hydrograph separation. The runoff-precipitation elasticity ("stream_elas") characterizes the inertia of change in mean runoff given a change in mean precipitation (Sankarasubramanian et al., 2001). For example, a value of 3 would indicate a change in runoff of 3% given a change in precipitation of 1%. High runoff-precipitation elasticity is especially present in the eastern part of the project area (Fig. 7f). The fraction of days without discharge (not shown, "zero_q_freq") may indicate strong infiltration (e.g. Danube Sinkhole; Hötzl, 1996), artificial water withdrawal, or ceasing baseflow.

The seasonality of runoff is expressed by the attribute "hfd_mean", which shows the number of days from the beginning of the hydrologic year (1 October) to the date when half annual of the runoff volume is reached (Court, 1962). Higher number of days in Fig. 7c can be explained primarily by water storage in form of snow (Fig. 6g) or glaciers (Fig. 8d). Variability in runoff (Fig. 7d, "slope_fdc") is expressed within LamaH by the slope of the flow duration curve between the log-transformed 33rd and 66th runoff percentiles (Sawicz et al., 2011). High values are indicative for high runoff variability over the year, which can be caused by seasonal water storage in the form of snow (Fig. 6g) or a strong response of runoff to precipitation (Yokoo and Sivapalan, 2011). Extreme runoff events such as high or low flow are described by indices representing mean frequency (Fig. 7g, "high_q_freq" / Fig. 7j, "low_q_freq"), duration (Fig. 7h, "high_q_dur" / Fig. 7k, "low_q_dur") and magnitude. The threshold for high flow (at least 9 times median daily discharge) is chosen according to Clausen and Biggs (2000), and that for low flow (max. 0.2 times median daily discharge) according to Olden and Poff (2003). The magnitudes of extreme flows are expressed by the 95th (high flow, Fig. 7i, "$Q_{95}$") and the 5th (low flow, Fig. 7l, "$Q_5$") runoff percentiles. The hydrological indices (Fig. 7) are spatially less smoothly distributed compared to the climatic indices (Fig. 6). The reasons might be the influence of the (non-) linear hydrological processes by locally heterogeneous catchment characteristics or uncertainties in runoff measurement (Addor et al., 2017a; Westerberg et al., 2016).

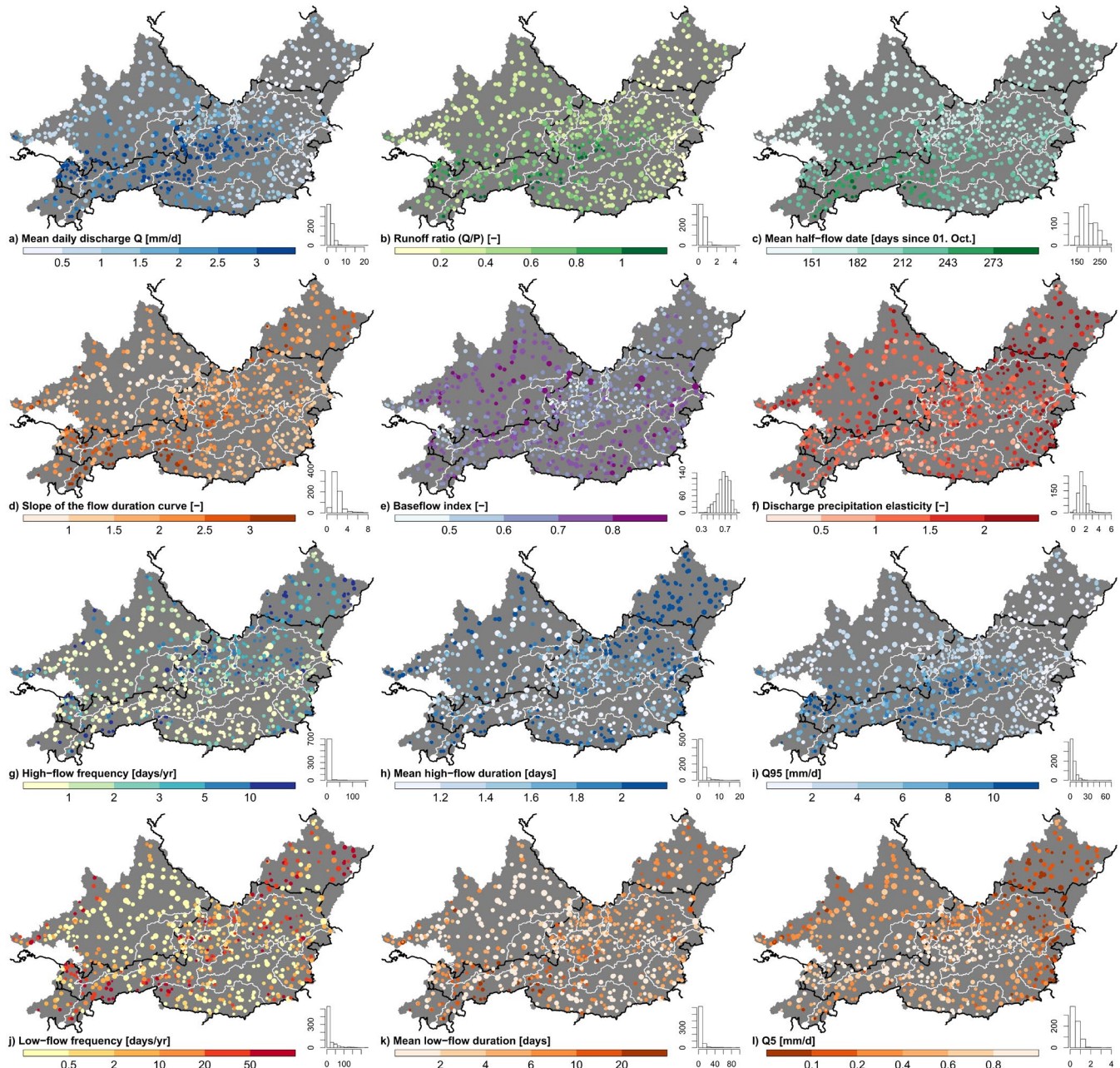

**Fig. 7: Spatial distribution of hydrological signatures.** Only gauges are plotted, which cover the period 1 October 1989 to 30 September 2009. The histograms indicate the number of gauges (out of 717) in each category. The size of the circles is proportional to the catchment area. © EuroGeographics for the administrative boundaries.

## 5.4 Land cover characteristics

All attributes concerning land class (Table A6) are based on the CORINE Land Cover (CLC) 2012 raster dataset featuring a grid-size of 100 m (CORINE, 2012). CORINE is an initiative of the European Environment Agency with the aim to record

land cover of the European territory with a 6-year update cycle. The basic technical specifications like 44 land classes, 25 hectares minimum mapping unit (MMU) for areal phenomena and 100 m minimum width for linear phenomena have not changed since the beginning, facilitating comparisons over the years (CORINE, 2012). It should be noted that an MMU of 25 hectares prevents mapping of very small scaled structures. Other limitations might be the variability of satellite image quality and contents, difficulties for setting up automatic conversion processes, the difference between human interpretation capacity and pixel-based classification (Bossard et al., 2000). However, the total reliability of the predecessor dataset CLC 2000 is 87.0 ± 0.8% according to a reinterpretation approach. The worst class-level reliability (< 70%) was found for sparse vegetation (CLC class 333) (Büttner and Maucha, 2006). The dominant land class within a basin delineation is derived by the majority of the intersecting raster centroids, while the fractions are derived by area share of the specific raster cells.

Agricultural land (Fig. 8a, "agr_fra" in Table A6) has high fractions in catchments with low mean slope (Fig. 5d, R = -0.89). The opposite occurs for fraction of bare areas (Fig. 8b, "bare_fra"), since the vegetation period is very short at high elevated terrain and a high terrain slope fosters gravitational erosion processes. Following the CAMELS datasets, no differentiation was made between deciduous and coniferous forests when calculating the forest share. The proportion of forest is highest in the central-eastern region of the project area (Fig. 8c, "forest_fra"), where agriculture and settlement are less prevalent and the mountains are often lower than the forest line. Catchments with a relatively high proportion of glaciers (Fig. 8d, "glac_fra") are mainly located in the western Eastern Alps. The influence of glaciers upon the hydrological regime is primarily apparent in the upper parts of the river regions Inn (region 3 in Fig. 1 and Table B1), Salzach (region 4) and Drava (region 18). High proportions of water surface (Fig. 8e, "lake_fra") can be explained by large lakes, which were mostly formed at the end of the last great ice age about 10 000 years ago (mainly in the Alpine foothills), or by large artificial water reservoirs (mainly in the Czech Republic). Catchments in the Vienna metropolitan area (eastern part of river region 10), as well as in the lower Rhine valley (northern part of river region 1) show quite high fractions of urban area (Fig. 8f, "urban_fra"). However, most catchments (about 74%) have less than 5% urban area.

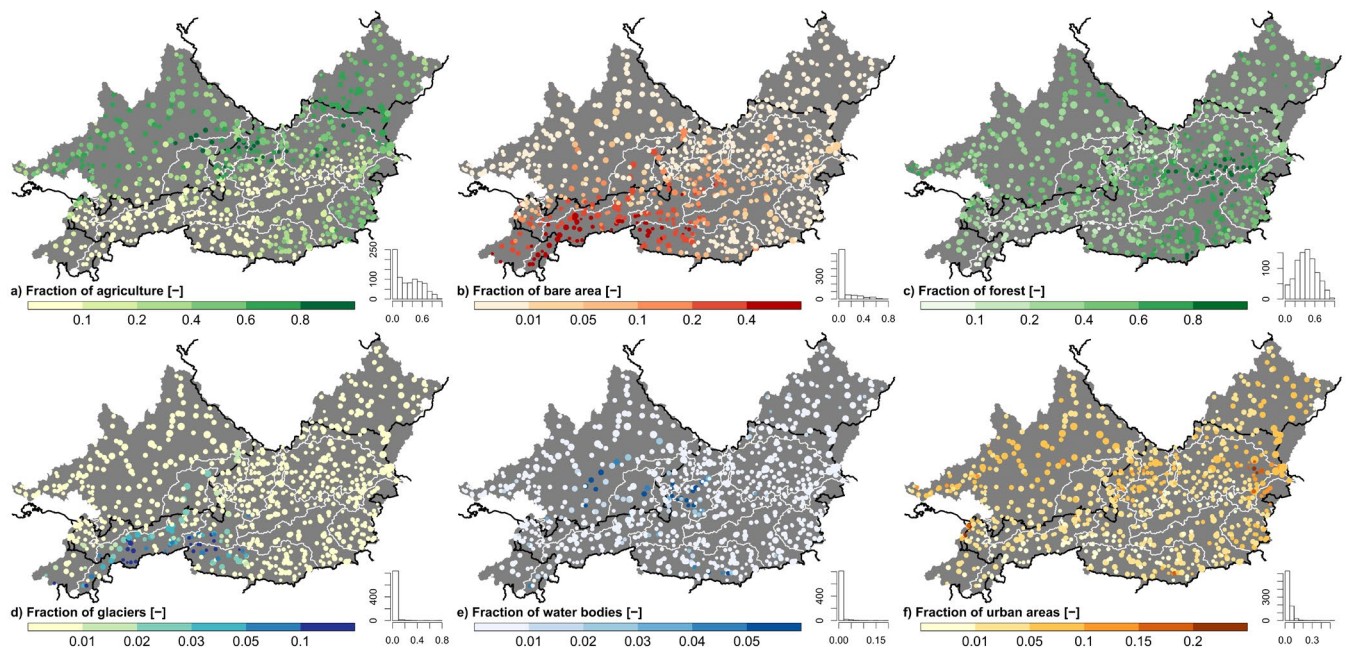

**Fig. 8: Spatial distribution of land class fractions representing the characteristics of the entire topographic catchment (basin delineation A, Fig. 2a). The histograms indicate the number of basins (out of 859) in each category. The size of the circles is proportional to the catchment area. © EuroGeographics for the administrative boundaries.**

## 5.5 Vegetation indices

We calculated 6 catchment attributes describing vegetation indices, which are based on Leaf Area Index (LAI), Normalized Difference Vegetation Index (NDVI), and Green Vegetation Fraction (GVF) (Table A7). All vegetation indices are based on long-term monthly means, using either the maximum, minimum, or difference between the maximum and minimum monthly means (based on 12 monthly means). Processing of the remote sensing datasets was done using the Google Earth Engine platform (GEEa, 2021; GEEb, 2021; Gorelick et al., 2017).

LAI represents vertical vegetation density and is defined as the sum of one-sided green leaf area per unit area for deciduous forests and half of the total needle area per unit area for coniferous forests. LAI was derived from the "MODIS MCD15A3H" dataset, which is a 4-day composition with 500 m grid resolution (Myneni et al., 2015). The maximum and minimum monthly means were calculated for the period 1 August 2002 to 1 January 2020 using a cloud filter. The maximum monthly mean of LAI (Fig. 9a, "lai_max" in Table A7) and the difference between maximum and minimum (Fig. 9d, "lai_diff") show a spatial correlation with the forest fraction (Fig. 8c, R = 0.76 respectively 0.75). $LAI_{diff}$ shows the same values as $LAI_{max}$ for large parts of the project area. Especially in regions characterized by a high proportion of coniferous forest the $LAI_{diff}$ should be smaller than $LAI_{max}$ due to the permanent green cover. Snow cover during the winter months could be a possible reason for the non-representative measurement of the minimum values of LAI.

NDVI is derived from the backscatter of 2 spectral bands and is widely used for remote sensing-based vegetation monitoring and classification (horizontal density, type, and physiological condition). The maximum and minimum monthly NDVI is based

on the "MODIS MOD09Q1" dataset with a temporal resolution of 8 days and a spatial resolution of 250 m (Vermote, 2015). The calculation was performed for the period 1 April 2000 to 1 January 2020, applying a filter on cloudy satellite images. A negative correlation is apparent between the NDVI$_{max}$ (Fig. 9b, "ndvi_max", R = -0.78) or the NDVI$_{min}$ (Fig. 9e, "ndvi_min", R = -0.84) and the mean catchment elevation (Fig. 5b).

GVF (Green Vegetation Fraction) indicates the fraction of soil that is covered by green vegetation and can be derived from the NDVI as follows in Eq. (1) (Broxton et al., 2014):

$$GVF = \frac{NDVI - NDVI_s}{NDVI_{c,v} - NDVI_s} \tag{1}$$

where NDVI represents the (maximum or minimum) monthly mean of NDVI, NDVI$_s$ the annual maximum NDVI of bare ground, and NDVI$_{c,v}$ the annual maximum of vegetated ground surface as a function of IGBP land class (Table 1 in Broxton et al., 2014). NDVI$_s$ was set to 0.09 in accordance to Broxton et al. (2014), while the spatial distribution of the IGBP land

classes was obtained from the "MODIS MCD12Q1" dataset of the year 2012 (Friedl and Sulla-Menashe, 2019). As the values for NDVI$_s$ and NDVI$_{c,v}$ were derived for a global scale and thus do not necessarily correspond to conditions in the project area, it is possible for GVF values to exceed the normal range between 0 and 1. In order to maintain consistency, we did not constrain the GVF to the normal range, however. The spatial distribution of GVF$_{max}$ (Fig. 9c, "gvf_max") shows similar spatial patterns to those of LAI$_{max}$ (R = 0.79) as well as NDVI$_{max}$ (R = 0.94), while GVF$_{diff}$ (Fig. 9f, "gfv_diff") tends to be higher in regions

with a higher fraction of precipitation falling as snow (Fig. 6g).

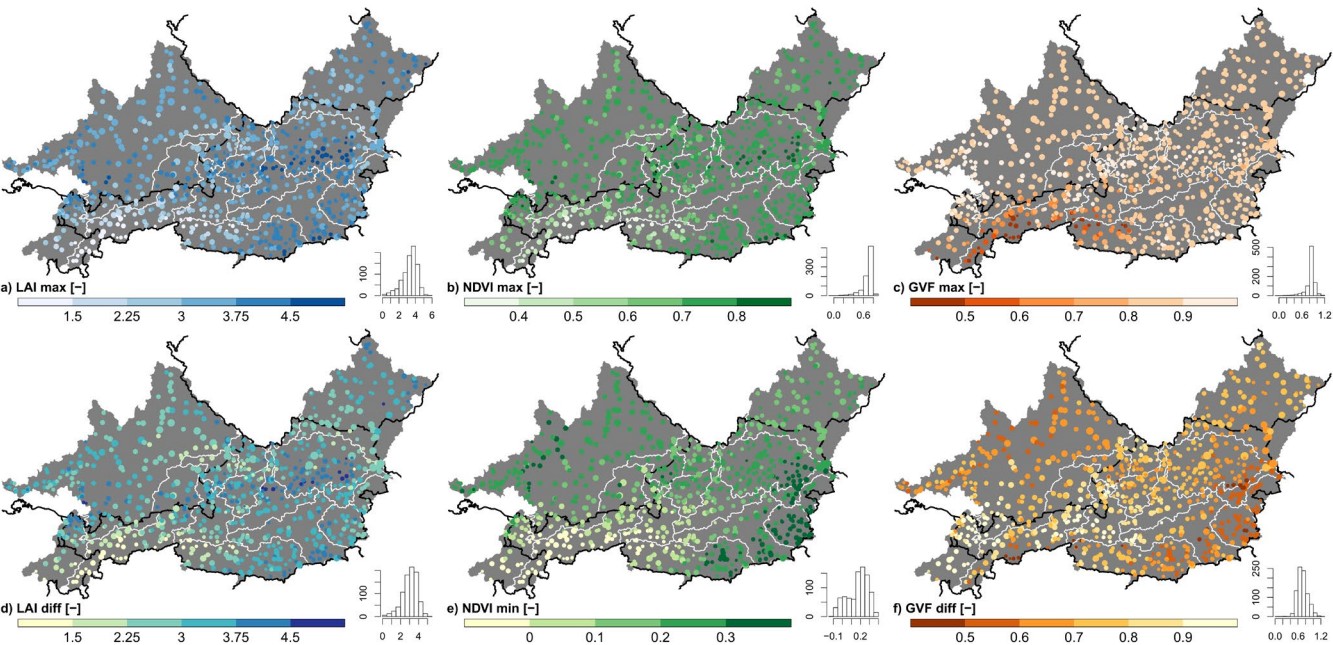

**Fig. 9: Spatial distribution of vegetation indices representing the characteristics of the entire topographic catchment (basin delineation A, Fig. 2a). The histograms indicate the number of basins (out of 859) in each category. The size of the circles is proportional to the catchment area. © EuroGeographics for the administrative boundaries.**

## 5.6 Soil characteristics

LamaH includes 10 attributes to characterize soil properties (Table A8), where 8 of them are derived from the 1 km grid sized "European Soil Database Derived Data" (ESDD; Hiederer, 2013a; 2013b). ESDD is based on the "European Soil Database" (ESD; Panagos et al., 2012; Panagos, 2006), while the maximum available soil water content (TAWC) in ESDD was calculated using pedotransfer functions (Hiederer, 2013a). ESDD provides soil attributes for a topsoil layer and a subsoil layer having the boundary at 30 cm soil depth. Values from these two layers were therefore aggregated, weighted by the available root depth ("root_dep" in Table A8), or in the case of TAWC summed up. The attribute describing the depth to bedrock "bdrk_dep" is based on the layer "average soil and sedimentary-deposit thickness" of the dataset "Global 1-km Gridded Thickness of Soil, Regolith, and Sedimentary Deposit Layers" (GGT; Pelletier et al., 2016). GGT has a spatial resolution of 30 arc seconds (approximately 1 km) and is derived from landform-specific models (for upland, lowland, slope, and valley floor) considering geomorphological principles, and incorporating data for topography, climate, and geology. Calibration and validation in GGT were performed using independent borehole profiles (Pelletier et al., 2016). The 3D Soil Hydraulic Database of Europe (3DSHD, Toth et al., 2017) dataset with a grid size of 250 m served as source for extracting the saturated hydraulic soil conductivity ("soil_condu"). 3DSHD was derived using pedotransfer functions (Toth et al., 2015) incorporating attributes from the SoilGrids250m dataset (SG250; Hengl et al., 2017), while SG250 is based on machine learning techniques including data from about 150 000 soil profiles as well as remote sensing data for climate, vegetation, geomorphology, and lithology (Hengl et al., 2017). Data within 3DSHD is provided for 7 soil layers, so a depth-weighted harmonic averaging was applied.

The provided soil attributes in LamaH may include large uncertainties and should therefore be considered with caution for several reasons. First, the soil attributes from ESD are mainly based on extrapolated observations of soil profiles and expert estimates (ESDB, 2004). Especially in the case of heterogeneous soil conditions and large distances between soil profiles the reliability of ESD dataset must be cautioned. Data from soil profiles are integrated in 3DSHD (Hengl et al., 2017; Toth et al., 2017) and the dataset of Pelletier et al. (2016) as well, but are rather used for calibration and validation. Toth et al. (2017) indicate increased unreliability for 3DSHD above 1000 m a.s.l. (about 24.2% of the project area is above 1000 m a.s.l.). Furthermore, the limitation of the soil depth at 1.5m in ESDD and 2.0 m in 3DSHD is another source of uncertainty (Boer-Euser et al., 2016). As a last point, it must be mentioned that much spatially distributed information is lost by aggregation to basin scale.

Depth to bedrock (Fig. 10a, "bedrk_dep" in Table A8) shows similar spatial patterns as mean catchment slope (Fig. 5d, R = -0.56), and mean elevation (Fig. 5b, R = -0.46). About 37 % of all 859 catchments have a mean depth to bedrock of more than 1.5 m. This depth represents the maximum root-available depth in ESDD (Fig. 10b, "root_dep"). The depth available for roots tends to be higher in Germany and the Czech Republic than in other regions. If this is an indication of different measurement methods across the countries is unclear. Low available rooting depths in Austria are, according to Fig. 10b, mainly present where the fraction of carbonate sedimentary rocks (Fig. 11c) or glaciers (Fig. 8d) is high. 40% of all catchments exhibit a mean organic soil content below 1%, while the highest organic contents are located in the southern German region (Fig. 10c,

"oc_fra"). Further interrelationships between the various grain size fractions and the dominating bedrock are recognizable: 1) A high proportion of sand (Fig. 10d, "sand_fra") is especially prevalent where the fraction of metamorphic bedrock is high (Fig. 11b, R = 0.47). 2) Moreover, the fraction of silt (Fig. 10e, "silt_fra") tends to be high on catchment level where a high fraction of carbonate sedimentary rock (Fig. 11c, R = 0.52) is present. 3) Finally, we can observe an increase in clay content (Fig. 10f, "clay_fra") with increasing proportion of mixed sedimentary rock (Fig. 11d, R = 0.47). Soil porosity (Fig. 10g, "soil_poros") shows similar spatial patterns compared to sand fraction (R = -0.79), while saturated hydraulic conductivity (Fig. 10h, "soil_condu") tends to increase with decreasing mean catchment elevation (Fig. 5b, R = -0.63). The available soil water depth (TAWC, "soil_tawc") was determined in ESDD by including water content at field capacity, gravel content and root-available depth (Hiederer, 2013a). That explains the high correlation of TAWC (Fig. 10i) with the root-available soil depth (Fig. 10b, R = 0.94).

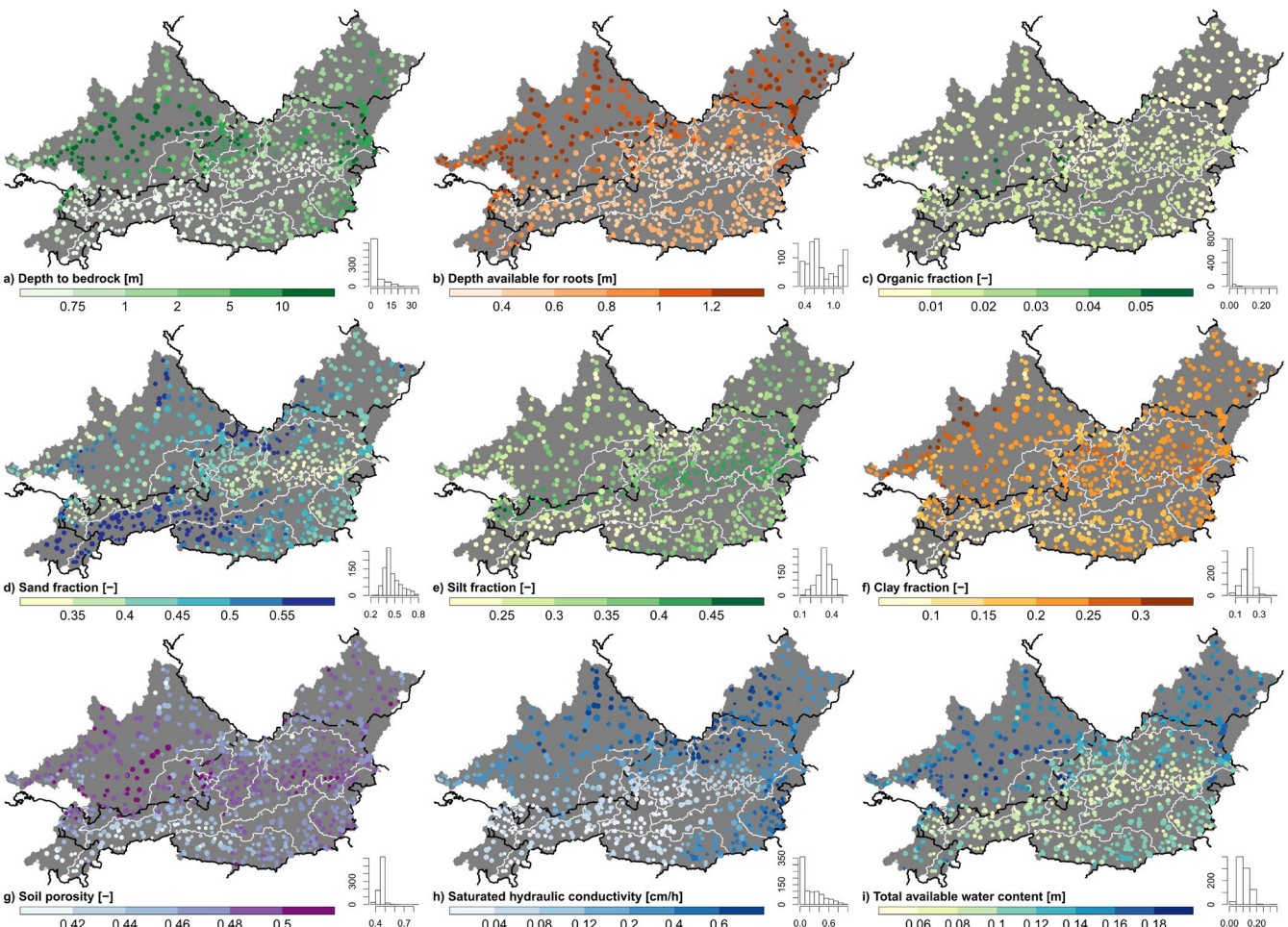

**Fig. 10: Spatial distribution of soil attributes representing the characteristics of the entire topographic catchment (basin delineation A, Fig. 2a). The histograms indicate the number of basins (out of 859) in each category. The size of the circles is proportional to the catchment area. © EuroGeographics for the administrative boundaries.**

## 5.7 Geologic characteristics

We used the datasets GLiM (Hartmann and Moosdorf, 2012; Global Lithological Map) and GLHYMPS (Gleeson et al., 2014; Global Hydrogeology Maps) for deriving 16 geologic attributes (Table A9). GLiM summarizes 92 regional geological maps in vector form and was used to extract the fractions of the different geological classes. GLiM offers 3 levels of detail, while the 1st level species the dominant lithologic class. The optional 2nd as well as 3rd level further specify, for example, the structure of the rock or local conditions (Hartmann and Moosdorf, 2012). For LamaH only the 1st level of GLiM was used, which contains 16 geological classes. The classes "evaporites", "no data" and " intermediate volcanic rocks" do not occur within the project area. The 3 most common dominant geologic classes (Fig. 11a, "gc_dom" in Table A9) across all 859 catchments are metamorphites (mt, 35.1%), carbonate sedimentary rocks (sc, 27.4%), and mixed sedimentary rocks (sm, 21.2%). Metamorphic rocks (Fig. 11b, "gc_mt_fra") are predominant along the northern border of the project area (Bohemian Massif), as well as in the more southern project area (Central Eastern Alps) and include mainly schist, gneiss and quartzite. From a hydrological point of view, the proportion of carbonate sedimentary rock is of particular interest, since a high fraction can be an indicator for karstic systems. High shares of carbonate sedimentary rocks are mainly found along the belt from the southwest to the central east of the project area (Northern Limestone Alps), the central southern border (Southern Limestone Alps) and the north-eastern border (Swabian Alb) (Fig. 11c, "gc_sc_fra"). The flysch and molasse zone (Alpine foothills and central parts of the German project area) is basically characterized by a high fraction of mixed sedimentary rocks (Fig. 11d, "gc_sm_fra"). Attributes concerning permeability and porosity of the lithologic bedrock were extracted from GLHYMPS. There is a high spatial correlation between GLHYMPS and GLiM, as geologic classes of GLiM served as a starting point for assigning hydraulic properties in GLHYMPS. Huscroft et al. (2018) declares that permeability in GLHYMPS is determined only for saturated conditions. GLHYMPS is only intended for regional-scale applications (i.e. spatial resolution greater than 5 km), as the influence of local heterogeneities such as fault zones can be neglected above this scale (Gleeson et al., 2014).

A high proportion of metamorphites or plutonites (mt, pa, pi in Fig. 11a) is commonly associated by low bedrock porosity (Fig. 11e, "geol_poros"). Catchments within the flysch and molasse zones in contrast exhibit relatively high porosity. High bedrock porosity is not necessarily followed by high subsurface permeability ("geol_perme"), yielding a much more inhomogeneous spatial pattern in Fig. 11f than in Fig. 11e. The reason may be rock structure (2nd stage of GLiM), which can have different impacts on permeability and porosity (Table 1 in Gleeson et al., 2014).

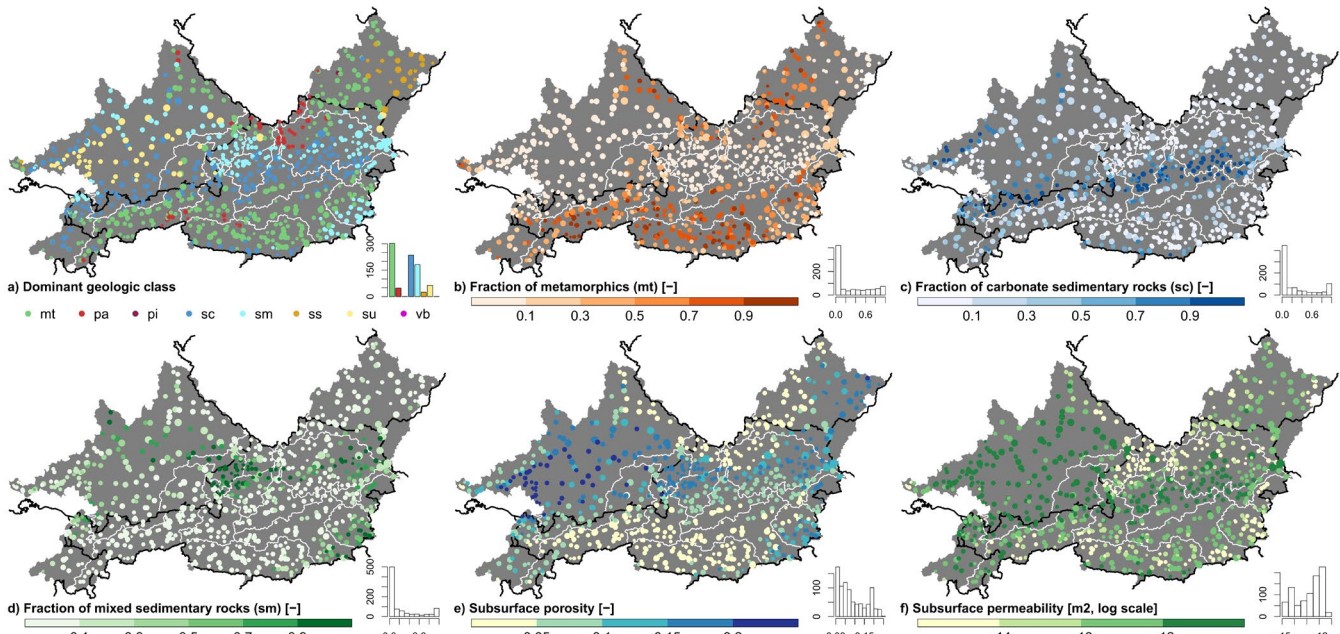

**Fig. 11:** Spatial distribution of geological attributes representing the characteristics of the entire topographic catchment (basin delineation A, Fig. 2a). The histograms indicate the number of basins (out of 859) in each category. The size of the circles is proportional to the catchment area. Classes in plot a): mt – Metamorphites, pa – Acid plutonic rocks, pi – Intermediate plutonic rocks, sc – Carbonate sedimentary rocks, sm – Mixed sedimentary rocks, ss – Siliciclastic sedimentary rocks, su – Unconsolidated sediments, vb – Basic volcanic rocks. © EuroGeographics for the administrative boundaries.

## 5.8 Information on (anthropogenic) impacts on runoff processes and measurements

We provide 4 attributes (Table 1) in order to simplify filtering and evaluation of time series of runoff gauges regarding any (anthropogenic) impact on runoff processes or its measurement. We have represented the diversity of (human) impact by 13 types of impact ("typimpact" in Table 1). The type of (human) impact on runoff or measurement was determined primarily from gauge-metadata declared by hydrographic services (BAFU, 2020; GKD, 2020; HZB, 2020; LUBW, 2020). Additionally, publicly available information, as well as manual aerial photo evaluations were used for determination. Typical types of human impact in the project area are large water reservoirs often associated with hydropower plants and cross-basin water transfers. The following types of influence were not classified because the necessary information is not consistently available, or only with great effort: 1) icing, especially at smaller rivers in winter; 2) variable channel profiles leading to inaccurate rating curves; 3) high groundwater flow in the area around the gauge; and 4) subsurface transboundary in- or outflows especially in highly karstified areas.

The hydrographs with hourly resolution in the months of January and July for the years 1990, 2005 and 2017 were additionally manually evaluated regarding systematic diurnal variations ("diur_art" / "diur_glac" in Table 1). Systematic fluctuations were further subdivided into those caused artificially (e.g. by storage power plants, power plants with swell operation or sewage treatment plants) and those caused naturally (snow or glacier melt). Summarizing the influences for every time series, the

degree of gauge impact ("degimpact") is determined mostly based on the type of impact and any systematic diurnal variations
(Table 2). Obviously, a gauge or catchment area can be characterized by several types of impact. In such cases, the highest
degree of impact was chosen. Geo-localization of the impacts is provided by the shapefile "Impacts.shp", which includes links
to the dam datasets GRanD ("GRAND_ID"; Lehner et al., 2011) and GOOD ("DAM_ID"; Mulligan et al., 2020) to ensure
fast access those attributes.

The spatial, as well as the frequency distribution of the degree of impact is shown in Fig. 12. 3.5% of 882 gauges are not
influenced (u), 48% show a low influence (l), 18.9% are moderately influenced (m) and 27% are strongly influenced (s), while
2.6% belong to class (x). Low influenced gauges are predominant in the northwest of the German project area, in the north of
the Austrian central region (river region 5, 6, 7, 8, 9, 10 in Fig. 1), in the east (river region 16), as well as in the south of Austria
(east of river region 18). Strongly influenced gauges are in contrast mainly prevalent, where large water reservoirs are in
operation for hydropower generation (primarily in the Alpine region) and for seasonal water balancing or flood protection
(primarily in the Czech Republic and in the north of the German project area). It should be noted that gauges located far
downstream of large reservoirs may still be strongly influenced by them.

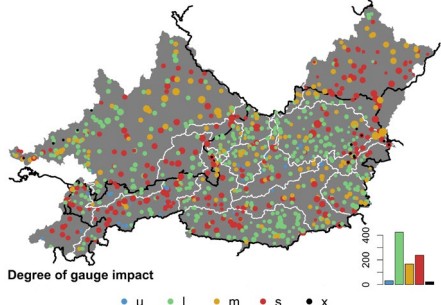

**Fig. 12: Degree of (anthropogenic) impact on gauges / catchments. The histogram indicates the number of gauges (out of 882) in each category. The size of the circles is proportional to the catchment area. Classes: u – no influence, l – low influence, m – moderate**
**influence, s – strong influence, x – not considered in basin delineation. © EuroGeographics for the administrative boundaries.**

**Table 1: Attributes for (anthropogenic) gauge and catchment interference.**

| Attribute | Description | Data source |
|---|---|---|
| typimpact | Type of gauge impact, categorized in 13 classes: | BAFU (2020), |
| | A – water reservoir with all-season water filling | CHMI (2020), |
| | B – flood retention reservoir | GKD (2020), |
| | C – lake with controllable outlet (e.g. weir) | HZB (2020), |
| | D – lake with unaffected outlet | LUBW (2020) |
| | E – water withdrawals | |
| | F – emergency outlet of water reservoir | |
| | G – extreme events are influenced / not properly measured | |
| | H – artificial channel | |

| | |
|---|---|
| | I – water intake (from neighbouring river or catchment) |
| | J – weed / vegetation near gauge |
| | K – fishing ponds |
| | L – high infiltration |
| | M – (karstic-) spring |
| diur_art | Unnatural systematic variations in hydrograph with hourly resolution, binary classification (0 - no, 1 - yes) | see above |
| diur_glac | Systematic fluctuations in the hydrograph with hourly resolution, which are caused by glacier melt, binary classification (0 - no, 1 - yes) | see above |
| degimpact | Degree of gauge impact, classes see Table 2. | see above |

**Table 2: Criteria for the different degrees of gauge impact.**

| degimpact | Criteria |
|---|---|
| u – no influence | If there is no obvious type of impact ("typimpact" in Table 1) and the gauge is located above populated areas. Artificial change of catchment size is less than 1% in case of impact type E (water withdrawals) or I (water intake) [a]. |
| l – low influence | Gauges located in or downstream of urban areas and without any type of impact (potential influence by undetected water withdrawals or stormwater drains) or type of impact is declared with D (lake with unaffected outlet) or J (weed / vegetation in gauge proximity). Artificial change of catchment size lies between 1 and 3% in case of impact type E or I [a]. |
| m – moderate influence | Type of impact is attributed with B (flood retention reservoir), C (lake with controllable outlet), F (emergency outlet of water reservoir), G (extreme events are influenced / not properly measured), K (fishing ponds) or L (high infiltration). An exception was made for 3 gauges in the upper Danube, which can be strongly (s) affected by full seepage during the summer months (Hötzl, 1996). Artificial change of catchment size is between 3 and 10% in case of impact type E or I [a]. |
| s – strong influence | Gauges with impact type A (water reservoir with all-season water filling) were assigned a strong (s) degree of impact in most cases. An exception was made in case of very large catchment areas (moderate degree of impact for selected Danube gauges). Artificial change of catchment size is more than 10% in case of impact type E or I [a]. Systematic diurnal variations of artificial origin ("diur_art" in Table 1). |
| x – not considered | Gauges, which 1) do not have a clearly assignable catchment area (e.g. gauges at artificial channels (impact type H) or below karstic springs), 2) are characterised by several time series (e.g. with or without consideration of mill channels; in this case it is possible that there are two gauges per point, one declared with "degimpact" = x and one with another degree of impact), and 3) have too many gaps (> 50%) in the |

| | time series. These gauges were subsequently assigned no meteorological time series or catchment attributes due to lack of basin delineation. |
|---|---|

[a] The hydrographic yearbook of Austria declares anthropogenic cross-basin water transfers by increasing or decreasing the natural catchment area of a gauge (BMLFUW, 2013). There is no information regarding artificial changes in catchment size for gauges outside Austria. Here, the degree of impact was additionally determined, and also at Austrian gauges influenced by other kinds of water withdrawal (river branches, diversions or irrigation), on the basis of publicly available information, as well as aerial photo analyses. We thereby mostly assigned a strong degree of impact (s), but in a few cases (e.g. withdrawals for drinking water) also a moderate degree (m).

## 6 Hydrological model

### 6.1 Model setup

Finally, we set up a conceptual hydrological model in order to check the inputs for plausibility and to be able to provide a baseline / benchmark model for further research. We applied the COSERO (COntinuous SEmi-distributed RunOff) model, which is a conceptual, semi-distributed hydrological model. It has a quite similar model structure as the well-known HBV model (Bergström, 1992). COSERO was developed in the 1990s at the University of Natural Resources and Life Sciences, Vienna, initially for runoff forecasting in alpine catchments in Austria (Nachtnebel et al., 1993). The model was also used in various hydrological studies in Austria (e.g. Nachtnebel and Fuchs, 2004; Eder et al., 2005; Kling and Nachtnebel, 2009a/b; Stanzel and Nachtnebel, 2010; Herrnegger et al., 2012, Kling et al., 2012, Frey and Holzmann, 2015; Herrnegger et al., 2015; Herrnegger et al., 2018; Klingler et al., 2020; Wesemann et al., 2018) and serves as a core for several operational discharge forecasting systems in Austria (e.g. Stanzel et al., 2008; Schulz et al., 2016; Wesemann et al., 2018). The performance of COSERO has been evaluated so far in different climates as well as in different spatiotemporal resolutions (e.g. Enzinger, 2009; Kling et al., 2015; Mehdi et al., 2021). COSERO incorporates interception, soil water storage, snow accumulation and melting (modified temperature-index approach, including log-normal distribution of snow-depth, cold-content of snow-pack, water holding capacity of snow-pack, refreezing of retained melt-water, settlement of snow-pack; see Frey and Holzmann, 2015), glacier melting, actual evapotranspiration (function of PET, snow sublimation, soil moisture and interception losses), division of runoff generation into different components (surface flow, interflow and baseflow), and routing through a cascade of (non-) linear reservoirs. Required inputs are time series for precipitation, air temperature and optionally PET as well as a parameter field including topology (Kling et al., 2015). Time series for PET can be derived model internally from the air temperature using the Thornthwaite approach (Thornthwaite and Mather, 1957).

Here, COSERO is applied with a lumped spatial discretization based on intermediate catchments (basin delineation B) and daily resolution. PET time series are derived internally following the Thornthwaite approach, since the PET time series from ERA5-Land are not included in LamaH (section 4.2). These derived PET time series are provided in addition to numerous other modelled fluxes within LamaH (Table C2). Artificial water reservoirs are not considered in COSERO. In contrast, cross-

basin water transfers using information from LamaH (see Table A11; "Crossbasin_water_transfers.csv"; "Impacts.shp") and

605 glaciers (if more than 10% area fraction) are accounted for. Calibration of 20 parameters (Table C1) was performed using the DDS algorithm (Tolson and Shoemaker, 2007) with a single-objective function (NSE, 100%) and 1 000 DDS iterations for the period 1 January 1982 to 30 September 2000. The year 1981 was used as a spin-up phase to enable system states to consolidate and reach an equilibrium. A (intermediate) basin was calibrated in an individual run, if the associated runoff gauge has observations at least since 1999. Otherwise (flag "fewobs" in supplementary text- and shapefiles is set to "1"), this basin was

610 treated as ungauged and calibrated together with the next downstream intermediate catchment, whose associated gauge has sufficiently long records. The results of those basins with no or to less runoff recordings in calibration phase are not evaluated (54 basins) and the runoff simulations are set to "-999" in the provided runoff simulations (Table C2). The period from 1 October 2000 to 30 September 2017 was used as validation phase.

## 6.2 Model results

We evaluate the model results using standard performance metrics NSE (Eq. 2; Gupta and Kling, 2011; Jain and Sudheer, 2008; Knoben et al., 2019; McCuen et al., 2006; Nash and Sutcliffe, 1970; Schaefli and Gupta, 2007) and percentual (p) long-term BIAS (Eq. 3).

$$NSE = 1 - \frac{\sum_t^n (Q_{sim,t} - Q_{obs,t})^2}{\sum_t^n (Q_{obs,t} - \overline{Q_{obs}})^2} \tag{2}$$

$$pBIAS = \frac{\overline{Q_{sim}} - \overline{Q_{obs}}}{\overline{Q_{obs}}} \times 100 \tag{3}$$

where $Q_{sim}$ represents the simulated and $Q_{obs}$ the gauged runoff. The dash above the variable indicates the arithmetic mean.
The NSE ranges from -9.26 (calibration) / -13.96 (validation) to 0.91 / 0.90 (Fig. 13b) with an area-unweighted median of 0.64

620 / 0.60. Inadequate model performance (Fig. 13a) can mostly be explained by: (i) cross-basin water transfers in karstified regions (see Fig. 11c), which is not accounted for in the model, (ii) a clear water surplus especially in eastern regions caused by overestimation in precipitation inputs or underestimation in evapotranspiration (Fig. 13c) or (iii) artificial structures, which were completed after the start of the calibration period (and thus were not specified in the artificial cross-basin water transfers). Rather good NSE values (> 0.6) can be observed primarily at: (i) gauges with large catchment area (Fig. 13a), (ii) rainfall-

625 dominated catchments (Fig. 6c) or (iii) gauges, which are not too strongly influenced by large water reservoirs.
The overall area-unweighted median pBIAS (Fig. 13c) is +6.1% in calibration and +4.4% in validation phase, which indicates either a precipitation surplus provided by ERA5-Land or underestimation of evapotranspiration. Herrnegger et al. (2012) show that the Thornthwaite approach tends to provide too low PET sums in alpine regions. Although air temperature is an important driver or proxy for evapotranspiration, other meteorological parameters, namely radiation, wind and relative humidity, are

630 equally or probably more important factors. This is especially the case, where lower air temperatures are present (especially in alpine regions) and other meteorological drivers for evapotranspiration, apart from temperature, become more important. The area-weighted mean of the PET-correction factor "ETSLPCOR" after calibration is 1.73 (with an upper boundary of 2.0,

see Table C1), which indicates a compensation of too low PET values in the calibration procedure. Considering that the long-term evapotranspiration totals of the model output ("ETAsum" in the supplied shapefile "Hyd_model.shp") seems quite plausible (e.g. compared to Fig. 20 in Herrnegger et al., 2012 or Map 3.3 in HAO, 2007), the reason for runoff surplus is likely a precipitation surplus in the ERA5-Land input. Klingler et al. (2020) show that CHIRPS Daily v2 (Funk et al., 2015) reflects long-term precipitation sums in the Mur catchment in the south of the Alps quite well. Fig. 4e in contrast indicates that ERA5-Land in general provides considerably higher precipitation sums compared to CHIRPS Daily v2. This, in combination with our restriction to simulate somewhat realistic ETA-fluxes, probably explains the many positive biases in the simulations of alpine catchments. The tight corset regarding ETA-fluxes, in combination with too high precipitation input, clearly leads to a lower model performance. Machine Learning approaches, with few exceptions (e.g. Hoedt et al., 2021), ignore these physical constraints and it is clear that higher model performance can be achieved, when ignoring the mass-balance or the realistic partitioning of precipitation in ETA and runoff.

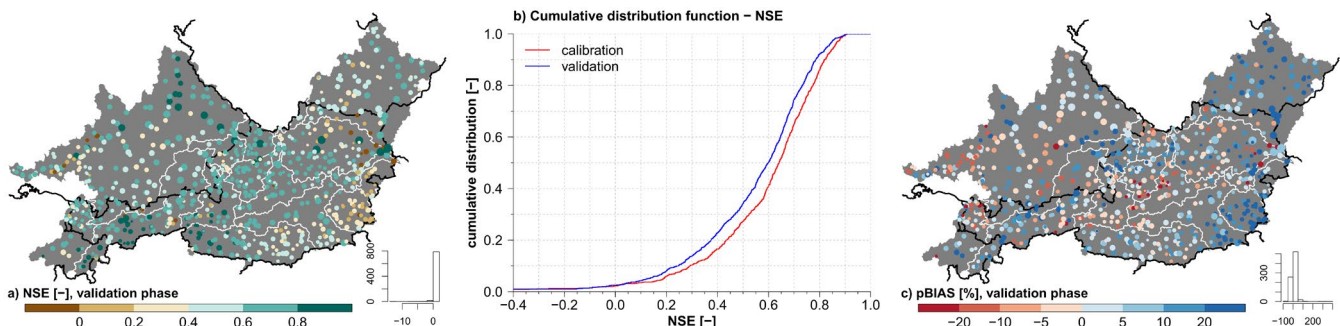

Fig. 13: Plot a) Spatial distribution of NSE in validation phase. Plot b) Cumulative distribution of NSE in calibration and validation phase. Plot c) Spatial distribution of percent bias in validation phase (positive values indicate a simulated surplus). The size of the circles in plot a) and c) is proportional to the catchment area. © EuroGeographics for the administrative boundaries.

Lastly, providing something like a disclaimer, it is important to stress that the simulation results stem from a large-scale model, which in this form has previously not been available. To our knowledge, no hydrological model existed, which (i) covers such a large domain in Central Europe in such detail (~170 000 km² in 9 countries divided in 859 sub-basins), (ii) uses so many discharge observations for calibration and validation, and (iii) which considers cross-basin water transfers. Although great care and love for detail was invested in the model setup of COSERO, it cannot be guaranteed that all local hydrometeorological features are represented and room for improvement probably remains. Consequently, this baseline model may locally exhibit significant deviations from real-world hydrologic conditions. This however generally remains a challenge for many large-scale hydrological models.

## 7 Summary and conclusions

Hydrological studies often require an extensive foundation of data. In large-scale or cross-national projects, it is therefore often laborious and time-consuming to collect the required data and then to homogenize the usually different formats, definitions

and conventions. Reasons are for instance the different organizational forms of the hydrographic authorities or communication
barriers. LamaH provides a unique, homogeneous data base for hydrological and other environmental sciences, that can overcome the mentioned barriers. Apart from the complete territory of Austria, LamaH includes all neighboring upstream areas of the rivers flowing through Austria as well. LamaH contains runoff time series as well as 15 meteorological time series (daily and hourly resolution) and over 60 attributes for 859 catchments. Additionally, simulations from a conceptual hydrological model provides a baseline for further investigations. Three basin delineations allow investigations with individual
catchments (as known from CAMELS) as well as within an interconnected river network considering intermediate catchments. It is clear that LamaH contains deficits and uncertainties due to the large number of data sources included. We however tried to consider and discuss most of these limitations.

Blöschl et al. (2019b) highlighted numerous open hydrological challenges, such as runoff prediction in ungauged basins (PUB). Methods based on machine learning show promising results for time series prediction (e.g. Kratzert et al., 2019a; 2019b;
Kratzert et al., 2018). However, uniformly structured "large-sample" datasets are helpful when applying these data-driven methods, because on the one hand the necessary preparatory work is drastically reduced and on the other hand the exchange or comparability of the modelling results is considerably facilitated. Given the scope of LamaH, we hope that this dataset will serve as a solid data base for further investigations in various fields of hydrology and adjacent fields of environmental science. The high variability in the data in combination with the interconnected river network as well as the high temporal resolution
of the time series could grant an improved understanding of processes in water transfer and storage, if appropriate methods are used.

**Data availability.** LamaH is freely available at https://doi.org/10.5281/zenodo.4525244 (Klingler et al., 2021). The dataset is basically divided into 7 parts including basin delineation A/B/C, gauges, stream network, hydrological model and appendix.
The first 4 parts mentioned contain shapefiles, various text files regarding the attributes as well as time series. The stream network is available with shapefiles, which contain numerous attributes. Various in- and outputs (e.g. parameter field, fluxes or evaluations) are provided for the hydrologic model. The entire folder structure, supplementary information regarding the time series, and required references are in the folder "Info". The runoff time series of the German federal states Bavaria and Baden-Württemberg are retrospective checked and updated by the hydrographic services. Therefore, it might be appropriate
to obtain more up-to-date runoff data from GKD (2020) or LUBW (2020). Please consider also the disclaimer stated at Zenodo.
**Code availability.** We have used R-Codes from Nans Addor (Addor, 2017b) for reproducing the climatic (Table A4) and hydrological (Table A5) indices as well as for creating the Figures 3, 5 to 13a/c. The color schemes in the plots are often based on ColorBrewer 2.0 (Brewer, 2021). Further relevant R- and Python-scripts for reproducing the dataset are available in the folder "G_appendix".
**Required additional references when using LamaH.** We ask kindly for compliance, as the agreement to citation was usually the claim for sharing the data. BAFU, 2020; CHMI, 2020; GKD, 2020; HZB, 2020; LUBW, 2020**;** BMLFUW, 2013; Broxton et al., 2014; CORINE, 2012; EEA, 2019; ESDB, 2004; Farr et al., 2007; Friedl and Sulla-Menashe, 2019; Gleeson et al., 2014;

HAO, 2007; Hartmann and Moosdorf, 2012; Hiederer, 2013a; Hiederer, 2013b; Linke et al., 2019; Muñoz Sabater et al., 2021; Muñoz Sabater, 2019a; Myneni et al., 2015; Pelletier et al., 2016; Toth et al., 2017; Trabucco and Zomer, 2019; Vermote 2015.

**Author contributions.** CK, KS, MH initiated the investigation and designed the study. CK requested the data base, processed the time series and computed the various attributes. The hydrological baseline model was set up by CK and MH while the manuscript was prepared by CK with contributions from all co-authors.

**Competing interests.** The authors declare that they have no conflict of interest.

**Financial support.** This work was in parts funded by the Austrian Science Fund FWF, project number P 31213.

**Acknowledgements.** We would like to thank the hydrographic offices from the (federal) states Austria, Baden-Württemberg, Bayern, Czech Republic and Switzerland for providing the runoff time series. Data processing was performed using the freely available software packages R (R Core Team, 2020), Python (Python Software Foundation, 2020), and QGIS (QGIS Development Team, 2020). Special thanks to all who have developed the numerous open source software, packages and extensions or who share their experiences in the numerous online forums. LamaH would not have been possible without the 705 institutions, working groups and individuals who worked, in some cases for several years, on the used open-source datasets.

**Review statement.** We would like to acknowledge the valuable inputs from the two reviewers Gemma Coxon and Mathis Messager. Their accurate and comprehensive reviews helped to significantly improve the paper and the dataset. Gratitude is also paid to the editor Lukas Gudmundsson as well as Daniel Klotz, who suggested to implement a hydrological baseline model. Lastly, we would like to thank Frederik Kratzert, who did a lot of testing and contributed some proposals for improving 710 the structure of LamaH.

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

## Appendix A

**Table A1: Gauge referred attributes.**

| Attribute | Description | Unit | References |
|---|---|---|---|
| ID | ID number, which is used for linking between all the different files | - | - |
| govnr | Official gauge number from the associated governments | - | BAFU (2020), CHMI (2020), GKD (2020), HZB (2020), LUBW (2020) |
| name | Official name of the runoff gauge | - | see above |
| river | Name of the belonging river | - | see above |

| | | | |
|---|---|---|---|
| area_gov | Catchment area obtained from the administration | km$^2$ | see above |
| elev | Elevation of the gauge`s zero point | m a.s.l. | see above |
| lon | Longitude in LAEA Europe grid (EPSG 3035) | m | see above |
| lat | Latitude in LAEA Europe (EPSG 3035) | m | see above |
| country | ISO 3166 ALPHA-3 code for country | - | see above |
| fedstate | Abbreviation for federal state; water management administration is executed in the given countries at the stage of federal states [a] | - | see above |
| region | River-based region [b] | - | see above |
| obsbeg_day | Start of daily runoff time series in the dataset | year | see above |
| obsbeg_hr | Start of continuous (hourly) runoff data recording | year | see above |
| obsend | End of continuous (hourly) runoff data recording; number 0 indicates an up-to-date recording; daily time series might last longer | year | see above |
| gaps_pre | Fraction of gaps in the raw hourly runoff time series | ‰ | see above |
| gaps_post | Fraction of gaps in the hourly runoff time series after our processing (linear interpolation, up to 6h) | ‰ | see above |
| area_ratio | Ratio between variable "area_calc" of basin delineation A (Table A3) and "area_gov" | - | BAFU (2020), CHMI (2020), GKD (2020), HZB (2020), LUBW (2020), HAO (2007), HydroATLAS (Linke et al., 2019) |
| nrs_euhyd | "OBJECT_ID" of the next river segment from stream network "EU-Hydro – River Network Database" [f] | - | EU-Hydro – River Network Database (EEA, 2019) |
| nrs_rivat | "HYRIV_ID" of the next river segment from stream network "RiverATLAS" [f] | - | HydroATLAS (Linke et al., 2019) |
| HIERARCHY | Gauge hierarchy [c] | - | see above |
| NEXTUPID | ID of the next upstream gauges (can be one or more), 0 indicates no upstream gauges [c] | - | see above |
| NEXTDOWNID | ID of the next downstream gauge (only one); 0 indicates no downstream gauge [c] | - | see above |
| dist_hup | Horizontal stream length from the most distant beginning of a watercourse within the basin to the gauge [d] | km | BAFU (2020), CHMI (2020), GKD (2020), HZB (2020), LUBW (2020), EU-Hydro – |

| | | | | |
|---|---|---|---|---|
| | | | | River Network Database (EEA, 2019) |
| dist_hdn | Horizontal stream length from the actual gauge to the next downstream gauge [c] | km | | see above |
| elev_diff | Elevation difference from the actual`s gauge zero point to the next downstream`s gauge zero point [c] | m | | see above |
| strm_slope | Slope of the actual gauge to the next downstream`s gauge; fraction of "elev_diff" and "dist_hdn" [c] | $m\ km^{-1}$ | | see above |
| ckhs | Declares if a timestep in the runoff time series was checked by the staff of the hydrological services or not; the specific scope of the check is not described [e] | 0 (False) or 1 (True) | | BAFU (2020), CHMI (2020), GKD (2020), HZB (2020), LUBW (2020) |
| qceq | QC flag which is set, if runoff remains equal in at least 10 consecutive time steps (daily or hourly); section 2.3 in Gudmundsson et al. (2018) [e] | 0 or 1 | | see above |
| qcol | QC flag which is set, if runoff value is classified as outlier; values which are declared as outliers can also be low/high flows in rare cases; section 2.3 in Gudmundsson et al. (2018) [e] | 0 or 1 | | see above |

[a] List of abbrev. for attribute fedstate: Austria (BLD – Burgenland, CRN – Carinthia, LAT – Lower Austria, SBG – Salzburg, STY – Styria, TYR – Tyrol, UAT – Upper Austria, VBG – Vorarlberg, VIE – Vienna); Germany (BAV – Bavaria, BWT – Baden-Württemberg); Switzerland (GRI – Grisons, STG – Saint Gallen); Liechtenstein (LIE - Liechtenstein); Czech Republic (OLM – Olomouc, SBO – South Bohemian, SMO – South Moravian, VYS – Vysočina, ZLN - Zlin). [b] List of abbrev. for attribute region: 1 – Rhine, 2 – Danube above Inn, 3 – Inn above Salzach, 4 – Salzach, 5 – Inn under Salzach, 6 – Danube between Inn and Traun, 7 – Traun, 8 – Danube between Traun and Enns, 9 – Enns, 10 – Danube between Enns and Morava, 11 – Vltava, 12 – Morava, 13 – Danube between Morava and Leitha, 14 – Leitha, 15 – Rabniz, 16 – Raab, 17 – Mur, 18 – Drava. [c] Only for basin delineation B and C. [d] Only for basin delineation A. [e] Visible in daily and hourly runoff time series. [f] End of the river segment (to which the attributes within the river network refer) can sometimes be rather far from the gauge. If a single river segment extended over several gauges, the ID of the river segment was only indicated at the most downstream gauge.

**Table A2: Meteorological variables from ERA5-Land dataset (Muñoz Sabater, 2019a).**

| Variable hourly | Daily aggregation | Description | | Unit |
|---|---|---|---|---|

| DOY | unchanged | Day of year | - |
|---|---|---|---|
| HOD | omitted | Hour of day | - |
| 2m_temp | max, mean, min | Air temperature at a height of 2 m above Earth surface | °C |
| 2m_dp_temp | max, mean, min | Dewpoint temperature at a height of 2 m above Earth surface | °C |
| 10m_wind_u | mean | Horizontal speed of air moving towards the east at a height of 10 m above Earth surface | m s$^{-1}$ |
| 10m_wind_v | mean | horizontal speed of air moving towards the north at a height of 10 m above Earth surface | m s$^{-1}$ |
| fcst_alb | mean | Forecast albedo, fraction of solar (shortwave) radiation reflected by Earth's surface (direct and diffuse) | - |
| lai_high_veg | mean | One-half of the total green leaf area per unit horizontal ground surface area for high vegetation type | - |
| lai_low_veg | mean | One-half of the total green leaf area per unit horizontal ground surface area for low vegetation type | - |
| swe | mean | Water equivalent of snow | mm |
| surf_net_solar_rad | max, mean | Amount of solar radiation (shortwave radiation) reaching the Earth`s surface (direct and diffuse) minus the amount reflected by the Earth's surface (governed by albedo); positive sign is indicator for radiation to the Earth | W m$^{-2}$ |
| surf_net_therm_rad | max, mean | Net thermal radiation at the Earth´s surface; positive sign is indicator for radiation from the Earth | W m$^{-2}$ |
| surf_press | mean | Surface pressure | Pa |
| total_et | sum | Total evapotranspiration, positive values indicate evapotranspiration, negative values condensation | mm |
| prec | sum | Total amount of precipitation (liquid and frozen) | mm |
| volsw_123 | mean | Fraction of water in top soil layer; 0 to 100 cm depth | m$^3$ m$^{-3}$ |
| volsw_4 | mean | Fraction of water in sub soil layer; 100 to 289 cm depth | m$^3$ m$^{-3}$ |

**Table A3: Topographic indices.**

| Attribute | Description | Unit | Data source |
|---|---|---|---|
| area_calc | Calculated basin area | km$^2$ | HAO (2007), HydroATLAS (Linke et al., 2019) |

| | | | |
|---|---|---|---|
| elev_mean | Mean catchment elevation [a] | m a.s.l. | NASA JPL SRTMGL1 V3 Digital Elevation 30m (Farr et al., 2007) |
| elev_med | Median catchment elevation [a] | m a.s.l. | see above |
| elev_std | Standard deviation of elevation in catchment [a] | m a.s.l. | see above |
| elev_ran | Range of catchment elevation (max. – min. elev.) [a] | m a.s.l. | see above |
| slope_mean | Mean catchment slope; Horn (1981) [a] | m km$^{-1}$ | see above |
| mvert_dist | Horizontal distance from the farthest point of the catchment to the belonging gauge (length axis) | km | BAFU (2020), CHMI (2020), GKD (2020), HZB (2020), LUBW (2020) |
| mvert_ang | Angle between North direction and connection from farthest point of catchment to belonging gauge (length axis); e.g. direction from north (farthest catchment point) to south (gauge): 180°, direction from east to west: 270° | degree | see above |
| elon_ratio | Elongation ratio $R_e$ after Schumm (1956); ratio between the diameter D of an equivalent circle and the area of the catchment area to its length L (mvert_dist), $$R_e = \frac{1}{L} \times \sqrt{\frac{4 \times A}{\pi}} = \frac{D}{L}$$ | - | see above |
| strm_dens | Stream density $D_F$, ratio of lengths of streams $L_F$ and the catchment area A (area_calc), $D_F = \frac{\sum L_F}{A}$ | km km$^{-2}$ | EU-Hydro – River Network Database (EEA, 2019) |

[a] Upscaling approach 2.


**Table A4: Climatic indices.**

| Attribute | Description | Unit | Data source |
|---|---|---|---|
| p_mean | Mean daily precipitation [a,d] | mm day$^{-1}$ | ERA5L (Muñoz Sabater, 2019a) |
| et0_mean | Mean daily reference evapotranspiration ET0 [b,e] | mm day$^{-1}$ | Global Aridity Index and Potential Evapotranspiration (ET0) Climate Database v2 (Trabucco and Zomer, 2019) |
| eta_mean | Mean daily total evapotranspiration [a,d] | mm day$^{-1}$ | ERA5L (Muñoz Sabater, 2019a) |

| | | | |
|---|---|---|---|
| arid_1 | Aridity, computed as the ratio of mean ET0 [b,e] (from Climate Database v2) and mean precipitation [a,d] (from ERA5-Land) | - | ERA5L (Muñoz Sabater, 2019a), Global Aridity Index and Potential Evapotranspiration (ET0) Climate Database v2 (Trabucco and Zomer, 2019) |
| arid_2 | Reciprocal value of aridity index from Climate Database v2 [b,e] | - | Global Aridity Index and Potential Evapotranspiration (ET0) Climate Database v2 (Trabucco and Zomer, 2019) |
| p_season | Seasonality and timing of precipitation (estimated using sine curves) to represent the annual precipitation cycles, positive (negative) values indicate that precipitation sums are higher during summer (winter) months; values close to 0 indicate uniform precipitation throughout the year; Eq. (14) in Woods (2009) [a,d] | - | see above |
| frac_snow | Fraction of precipitation falling as snow, i.e. falling on days with mean temperature below 0 °C [a,d] | - | see above |
| hi_prec_fr | Frequency of high precipitation days ($\geq 5$ times mean daily precipitation) [a,d] | days yr$^{-1}$ | see above |
| hi_prec_du | Mean duration of high precipitation events (number of consecutive days with $\geq 5$ times mean daily precipitation) [a,d] | days | see above |
| hi_prec_ti | Season during which most high precipitation days ($\geq 5$ times mean daily precipitation) occur [a,d] | season [c] | see above |
| lo_prec_fr | Frequency of dry days ($< 1$ mm day$^{-1}$ precipitation) [a,d] | days yr$^{-1}$ | see above |
| lo_prec_du | Mean duration of dry periods (number of consecutive days with $< 1$ mm day$^{-1}$ precipitation) [a,d] | days | see above |
| lo_prec_ti | Season during which most dry days ($< 1$ mm day$^{-1}$ precipitation) occur [a,d] | season [c] | see above |

[a] Period 1 October 1989 to 30 September 2009. [b] Period 1970 to 2000. [c] List of abbrev. for season: djf – December/January/February, mam – March/April/May, jja – June/July/August, son – September/October/November. [d] Upscaling approach 1. [e] Upscaling approach 2.


**Table A5: Hydrological signatures.**

| Attribute | Description | Unit | Data source |
|---|---|---|---|
| q_mean | Mean daily runoff [a,c,d] | mm day$^{-1}$ | BAFU (2020), CHMI (2020), GKD (2020), HZB (2020), LUBW (2020) |
| runoff_ratio | Runoff ratio, computed as the ratio of mean daily runoff and mean daily precipitation [a,b,c,d] | - | BAFU (2020), CHMI (2020), GKD (2020), HZB (2020), LUBW (2020), ERA5L (Muñoz Sabater, 2019a) |
| stream_elas | Runoff-precipitation elasticity, i.e. the sensitivity of runoff to changes in precipitation at the annual timescale, using the mean daily runoff as reference; Eq. (7) in Sankarasubramanian et al. (2001), with the last element being $\bar{P} / \bar{Q}$ not $\bar{Q} / \bar{P}$ [a,c,d] | - | BAFU (2020), CHMI (2020), GKD (2020), HZB (2020), LUBW (2020) |
| slope_fdc | Slope of the flow duration curve between the log-transformed 33rd and 66$^{th}$ runoff percentiles; Eq. (3) in Sawicz et al. (2011) [a,c,d] | - | see above |
| baseflow_index _ladson | Baseflow index, computed as the ratio of mean daily baseflow and mean daily discharge; hydrograph separation is performed using the Ladson et al. (2013) digital filter [a,c,d] | - | see above |
| baseflow_index _lfstat | equal than above, but hydrograph separation is performed using a package from Tallaksen and Van Lanen (2004) [a,c,d] | - | see above |
| hfd_mean | Mean half-flow date, i.e. number of days since the beginning of the hydrological year (1 October) on which the accumulated runoff reaches half of the annual volume; Court (1962) [a,c,d] | days since 1 October | see above |
| Q5 | 5% flow quantile (low flow) [a,c,d] | mm day$^{-1}$ | see above |
| Q95 | 95% flow quantile (high flow) [a,c,d] | mm day$^{-1}$ | see above |
| high_q_freq | Frequency of high-flow days (> 9 times median daily flow); Clausen and Biggs (2000), Table 2 in Westerberg and McMillan (2015) [a,c,d] | days yr$^{-1}$ | see above |

| | | | |
|---|---|---|---|
| high_q_dur | Mean duration of high-flow events (number of consecutive days > 9 times median daily flow); Clausen and Biggs (2000), Table 2 in Westerberg and McMillan (2015) [a,c,d] | days | see above |
| low_q_freq | Frequency of low-flow days (< 0.2 times mean daily flow); Olden and Poff (2003), Table 2 in Westerberg and McMillan (2015) [a,c,d] | days yr$^{-1}$ | see above |
| low_q_dur | Mean duration of low-flow events (number of consecutive days < 0.2 times mean daily flow); Olden and Poff (2003), Table 2 in Westerberg and McMillan (2015) [a,c,d] | days | see above |
| zero_q_freq | Percentage of days with no discharge [a,c,d] | % | see above |

[a] Period 1 October 1989 to 30 September 2009 and additionally from the first 1 October in the time series after 1981 to 30 September 2017. [b] Upscaling approach 1. [c] No values if there are more than 5% gaps in the calculation period. [d] No values for basins / gauges which are attributed with degimpact = "x".


**Table A6: Land cover characteristics.**

| Attribute | Description | Unit | Data source |
|---|---|---|---|
| lc_dom | 3-digit short code of dominant land cover class [a,b] | - | CORINE (2012) |
| agr_fra | Fraction of agricultural areas (all CLC classes starting with number 2) [a,b] | - | see above |
| bare_fra | Fraction of bare areas (CLC classes 332, 333) [a,b] | - | see above |
| forest_fra | Fraction of forest areas (CLC classes 311, 312, 313) [a,b] | - | see above |
| glac_fra | Fraction of glaciers (CLC class 335) [a,b] | - | see above |
| lake_fra | Fraction of natural or artificial water bodies with all-season water filling (CLC class 512) [a,b] | - | see above |
| urban_fra | Fraction of areas mainly occupied by buildings including their connected areas (CLC classes 111, 112, 121, 122, 123, 124) [a,b] | - | see above |

[a] Upscaling approach 2. [b] Land class nomenclature is listed in the folder "G_appendix".

**Table A7: Vegetation indices.**

| Attribute | Description | Unit | Data source |
|---|---|---|---|

| | | | |
|---|---|---|---|
| lai_max | Maximum monthly mean of one-sided leaf area index (based on 12 monthly means) [a] | - | MODIS MCD15A3H (Myneni et al., 2015) |
| lai_diff | Difference between maximum and minimum monthly mean of one-sided leaf area index (based on 12 monthly means) [a] | - | see above |
| ndvi_max | Maximum monthly mean of NDVI (based on 12 monthly means) [a] | - | MODIS MOD09Q1 (Vermote, 2015) |
| ndvi_min | Minimum monthly mean of NDVI (based on 12 monthly means) [a] | - | see above |
| gvf_max | Maximum monthly mean of the green vegetation fraction (based on 12 monthly means) [a] | - | MODIS MOD09Q1 (Vermote, 2015), MODIS MCD12Q1 (Friedl and Sulla-Menashe, 2019) |
| gvf_diff | Difference between the maximum and minimum monthly mean of the green vegetation fraction (based on 12 monthly means) [a] | - | see above |

[a] Upscaling approach 2.

**Table A8: Soil characteristics.**

| Attribute | Description | Unit | Data source |
|---|---|---|---|
| bedrk_dep | Depth to bedrock; maximum is 50 m [c] | m | Pelletier et al. (2016) |
| root_dep | Depth available for roots; maximum is 1.5 m [a,c] | m | European Soil Database Derived data (Hiederer, 2013a/b) |
| soil_poros | Total soil porosity [a,b,c] | - | see above |
| soil_condu | Saturated hydraulic conductivity; maximum is 2 m [a,b,c] | cm hr$^{-1}$ | 3D Soil Hydraulic Database of Europe (Toth et al., 2017) |
| soil_tawc | Total available water content (between field capacity and permanent wilting point) [a,b,c] | m | European Soil Database Derived data (Hiederer, 2013a/b) |
| sand_fra | Sand fraction (of soil material < 2 mm) [a,b,c] | - | see above |
| silt_fra | Silt fraction (of soil material < 2 mm) [a,b,c] | - | see above |
| clay_fra | Clay fraction (of soil material < 2 mm) [a,b,c] | - | see above |
| grav_fra | Fraction of gravel (of overall soil) [a,b,c] | - | see above |
| oc_fra | Fraction of organic material (of overall soil) [a,b,c] | - | see above |

[a] Areas marked as water or bedrock were excluded from calculation. [b] Aggregation weighted by depth over the different soil layers. [c] Upscaling approach 2.


**Table A9: Geologic attributes.**

| Attribute | Description | Unit | Data source |
|---|---|---|---|
| gc_dom | Dominant geologic class [a] | - | GLiM (Hartmann and Moosdorf, 2012) |
| gc_ig_fra | Fraction of "ice and glacier" (ig) [a] | - | see above |
| gc_mt_fra | Fraction of "metamorphites" (mt) [a] | - | see above |
| gc_pa_fra | Fraction of "acid plutonic rocks" (pa) [a] | - | see above |
| gc_pb_fra | Fraction of "basic plutonic rocks" (pa) [a] | - | see above |
| gc_pi_fra | Fraction of "intermediate plutonic rocks" (pi) [a] | - | see above |
| gc_py_fra | Fraction of "pyroclastics" (py) [a] | - | see above |
| gc_sc_fra | Fraction of "carbonate sedimentary rocks" (sc) [a] | - | see above |
| gc_sm_fra | Fraction of "mixed sedimentary rocks" (sm) [a] | - | see above |
| gc_ss_fra | Fraction of "siliciclastic sedimentary rocks" (ss) [a] | - | see above |
| gc_su_fra | Fraction of "unconsolidated sediments" (su) [a] | - | see above |
| gc_va_fra | Fraction of "acid volcanic rocks" (va) [a] | - | see above |
| gc_vb_fra | Fraction of "basic volcanic rocks" (vb) [a] | - | see above |
| gc_wb_fra | Fraction of "water bodies" (wb) [a] | - | see above |
| geol_perme | Subsurface permeability (log10) [b] | - | GLHYMPS (Gleeson et al., 2014) |
| peol_poros | Subsurface porosity [b] | - | see above |

[a] Upscaling approach 1. [b] Upscaling approach 2.

**Table A10: Attributes in the accompanying file "Water_balance.csv" [e]**

| Attribute | Description | Unit | Data source |
|---|---|---|---|
| P | Mean precipitation of ERA5-Land [a,b] | mm d$^{-1}$ | ERA5L (Muñoz Sabater, 2019a) |
| PCRPS | Mean precipitation of CHIPRS Daily v2.0 [a,b] | mm d$^{-1}$ | CHIRPS Daily v2.0 (Funk et at., 2015) |
| PERA5 | Mean precipitation of ERA5 [a,b] | mm d$^{-1}$ | ERA5 (Hersbach et al., 2020) |
| PMSW | Mean precipitation of MSWEP v2.2 [a,b] | mm d$^{-1}$ | MSWEP v2.2 (Beck et al., 2017; 2019) |

| PET | Mean PET of ERA5-Land [a,b] | mm d$^{-1}$ | ERA5L (Muñoz Sabater, 2019a) |
|---|---|---|---|
| ET0 | Mean ET0 of Climate Database v2 [a,c] | mm d$^{-1}$ | Global Aridity Index and Potential Evapotranspiration (ET0) Climate Database v2 (Trabucco and Zomer, 2019) |
| ETA | Mean actual evapotranspiration of ERA5-Land [a,b] | mm d$^{-1}$ | ERA5L (Muñoz Sabater, 2019a) |
| Q1 | Mean runoff height [b,f,h] | mm d$^{-1}$ | BAFU (2020), CHMI (2020), GKD (2020), HZB (2020), LUBW (2020) |
| Q2 | Mean runoff height [d,g,h] | mm d$^{-1}$ | see above |

[a] Computed for basin delineation A. [b] Period 1 October 1989 to 30 September 2009. [c] Period 1970 to 2000. [d] Period from the first 1 October in the time series after 1981 to 30 September 2017. [e] No values, where corresponding gauge is attributed with "degimpact" = x (Table 1). [f] No values for gauges, whose time series do not cover the full period. [g] Values only if at least 5 years of observations are available. [h] No values if there are more than 5% gaps in the calculation period.

**Table A11: Attributes in the accompanying file "Crossbasin_water_transfers.csv" [a]**

| Attribute | Description | Unit |
|---|---|---|
| ID | ID of sourcing basin | - |
| To_ID | ID of receiving basin | - |
| cross_region | Flag if river region (Fig. 1, Table B1) is crossed (0 - no, 1 - yes) | - |
| lower_thres | Lower threshold for water cross-basin transfer | m$^3$ s$^{-1}$ |
| upper_thres | Upper threshold for water cross-basin transfer | m$^3$ s$^{-1}$ |
| estimated | Flag if "lower_thres" and "lower_thres" are estimated (1) due to lack of publicly available information; otherwise this information is taken from governmental registers, system specifications, etc. (0) | - |
| model | Flag if the cross-basin transfer is implemented in the hydrological model COSERO (0 - no, 1 - yes); COSERO can only transfer water from a source basin to one single receiving basin | - |

[a] Derived for basin delineation B.

**Appendix B**

**Table B1: Overview of the river regions.**

| No. | Name of river region | Area [km²] [a] | Important headwaters and tributaries |
|---|---|---|---|
| 1 | Rhine | 7 610 | Anterior Rhine, Posterior Rhine, Plessur, Landquart, Ill, Frutz |
| 2 | Danube above Inn | 49 942 | Iller, Brenz, Wörnitz, Lech, Altmühl, Naab, Regen, Isar, Ilz, Vils |
| 3 | Inn above Salzach | 15 249 | Sanna, Ötztaler Ache, Sill, Ziller, Brandenberger Ache, Brixentaler Ache, Mangfall, Alz |
| 4 | Salzach | 6 684 | Krimmler Ache, Fuscher Ache, Gasteiner Ache, Großarlbach, Lammer, Saalach |
| 5 | Inn under Salzach | 4 060 | Mattig, Mühlheimer Ache, Rott, Pram |
| 6 | Danube between Inn and Traun | 2 870 | Erlau, Große Mühl, Innbach |
| 7 | Traun | 3 851 | Ischl, Ager, Krems |
| 8 | Danube between Traun and Enns | 342 | Gusen |
| 9 | Enns | 5 997 | Salza, Steyr |
| 10 | Danube between Enns and Morava | 15 492 | Aist, Naarn, Ybbs, Erlauf, Melk, Pielach, Kamp, Schwechat, Fischa |
| 11 | Vltava | 763 | Maltsch |
| 12 | Morava | 25 688 | Bečva, Thaya |
| 13 | Danube between Morava and Leitha | 1 073 | - |
| 14 | Leitha [b] | 2 118 | Schwarza |
| 15 | Rabnitz [b] | 777 | Einser-Kanal |
| 16 | Raab [b] | 4 416 | Lafnitz, Pinka |
| 17 | Mur [b] | 9 908 | Taurach, Pöls, Liesing, Mürz, Kainach, Sulm |
| 18 | Drava [b] | 12 000 | Isel, Möll, Lieser, Gail, Gurk, Lavant |

[a] Area which is covered by basin delineation B. [b] River joins the Danube outside the project area in Hungary / Croatia.

**Appendix C**

**Table C1: Calibrated parameters in the hydrological model COSERO.**

| Parameter | Description | Default value | Lower constraint | Upper constraint | Unit |
|---|---|---|---|---|---|
| | | | | | |

| | | | | | |
|---|---|---|---|---|---|
| BETA | Parameter to compute runoff generation as a function of soil moisture | 4.5 | 0.1 | 10 | - |
| CTMAX | Maximum snow melt factor on June 21 | 2 | 1 (5 [b]) | 7 | mm °C$^{-1}$ d$^{-1}$ |
| CTMIN | Minimum snow melt factor on Dec 21 | 5 | 1 | 7 | mm °C$^{-1}$ d$^{-1}$ |
| QDIV_RATIO [c] | Ratio of the water amount which is transferred to the receiving basin and the water amount which flows to the intake in the source basin | 0.5 | 0 | 0.9 | - |
| ETSLPCOR | Factor to correct PET | 1 | 0.5 | 2 | - |
| FKFAK | Factor to compute ETA from PET as a function of soil moisture | 0.7 | 0.1 | 1 | - |
| H1 | Outlet level of reservoir for simulating surface flow | 2 | 1 | 20 | mm |
| H2 | Outlet level of reservoir for simulating interflow | 10 | 0 | 50 | mm |
| KBF | Recession constant for simulating outflow from the soil module with a linear reservoir | 3 000 | 1 000 | 10 000 | h |
| M | Storage capacity of the soil | [a] | 10 | 500 | mm |
| NVAR | Variance for distributing new snowfall with a log-normal distribution | 1.5 | 0 | 10 (5 [b]) | - |
| RAINTRT | Transition temperature above which precipitation is pure rain | 3 | 0 | 4 | °C |
| SNOWTRT | Transition temperature below which precipitation is pure snow | 0.5 | -2 | 2 | °C |
| TAB1 | Recession constant for simulating surface flow | 50 | 1 | 200 | h |
| TAB2 | Recession constant for simulating interflow | 250 | 1 | 500 | h |
| TAB3 | Recession constant for simulating baseflow | 5 000 | 10 | 10 000 | h |
| TAB4 | Recession constant for simulating routing | 1 | 0.05 | 10 | h |
| TAB5 | Additional time shift for simulating routing | 1 | 0.05 | 20 | h |
| TVS1 | Recession constant for simulating percolation from the surface flow module | 100 | 1 | 400 | h |
| TVS2 | Constant for simulating percolation from the inter flow module | 500 | 1 | 1 000 | h |

[a] parameter "soil_tawc" (Table A8). [b] for basins with a mean catchment elevation ("elev_mean" in Table A3) of at least 2 000 m a.s.l. [c] calibrated only if there is a cross-basin water transfer (see file "Crossbasin_water_transfers.csv", Table A11), otherwise parameter is set to 0.

**Table C2: Provided input / output time series from the hydrological model COSERO.**

| Variable | Description | Unit | References |
|---|---|---|---|
| YYYY | Year | - | - |
| MM | Month | - | - |
| DD | Day | - | - |
| P_ | Precipitation sum [a] | mm | ERA5L (Muñoz Sabater, 2019a) |
| T_ | Mean air temperature [a] | °C | ERA5L (Muñoz Sabater, 2019a) |
| PET_ | Sum of potential evapotranspiration (PET) following Thornthwaite [a] | mm | Thornthwaite and Mather (1957) |
| ETA_ | Sum of actual (total) evapotranspiration (ETA) [a] | mm | - |
| BW0_ | Water level of soil storage [a] | mm | - |
| BW3_ | Water level of groundwater storage [a] | mm | - |
| SWW_ | Snow water equivalent [a] | mm | - |
| Qsim_ | Simulated runoff [a,b] | $m^3\ s^{-1}$ | - |
| Qobs | Gauged runoff | $m^3\ s^{-1}$ | BAFU (2020), CHMI (2020), GKD (2020), HZB (2020), LUBW (2020) |

[a] Suffix "_A" indicates that aggregation was done for basin delineation A (full upstream topographic catchment area), while suffix "_B" indicates aggregation for basin delineation B (intermediate catchments). [b] Set to "-999", if basin was calibrated together with the next downstream basin due to less runoff observations in calibration period.