# Peer review of "LamaH-CE | Large-Sample Data for Hydrology and Environmental Sciences for Central Europe"

_Earth System Science Data, 2021_

## Referee Comment (RC2)

**Review of 'LamaH | Large-Sample Data for Hydrology and Environmental Sciences for Central Europe' by Klinger et al. Submitted to Earth System Science Data.**

**Gemma Coxon**

This paper describes the development of a large sample hydrology dataset for Central Europe (predominantly Austria). It compiles data across 859 gauges providing meteorological and hydrological timeseries as well as a vast array of catchment attributes covering climate, hydrology, land use, geology, vegetation and human influences. The timeseries are provided at both hourly and daily timescales.

The authors need to be congratulated on compiling a fantastic dataset that will be of great value to the hydrological community. It is a huge effort and it is very evident that a lot of care and thought has gone into the dataset. There is extensive discussion of the data sources and processing steps in the paper. There is good consideration of uncertainties and the figures are generally nicely presented.

I recommend publication of the dataset and paper, but the authors have some work to clarify the basin delineation and code availability. The paper is mostly well written but needs a really thorough proofread with quite a few sentences that are unclear (see list of technical corrections but this is not exhaustive!).

**Main comments**

*Basin delineation and 'headwater' catchments.* The delineation of the catchment boundaries is a key feature of the dataset but currently is not clear. I suggest the following:
- *Terminology* needs to be much clearer. What do you mean by 'orographic catchment' (is this a commonly used term? Do you mean topographic catchment?). I don't believe you are using the term 'headwater catchment' correctly - I interpret headwater catchments as low order catchments found in the upper reaches of river basins. In which case statements like 'In contrast, however, LamaH does not only consider headwater basins' on L18 are not correct as CAMELS datasets also do not only consider headwater basins.
- *Data source* of the catchment boundaries – why did you use catchment boundaries from two products? It is not clear how the catchment boundaries are combined. L114 'As aggregation areas….' This sentence doesn't make sense and needs rewriting.
- *Figure 2.* I find this figure a little unclear and the figure caption is currently very long. It may be worth simplifying the figure and thinking about moving some of the explanatory text in the figure caption to the main text in the paper.

*Potential evapotranspiration.* The analysis in Section 4.2 is really interesting and an important addition to the paper. I understand the authors decision to not include the PET data of ERA-5,

but given the importance of this variable as forcing data for a large amount of hydrological models (particularly conceptual lumped hydrological models), it seems a shame not to include it as a variable. It is not entirely clear how you would derive PET from the reference evapotranspiration provided. Were other global PET products considered?

*Code availability.* Reproducibility for these large-sample datasets is key. The authors should consider making their code available alongside the dataset. I would also recommend a code availability statement to make clear the code that was used in the paper. For example, I believe you used Nans' code to calculate the hydrologic and climatic catchment attributes and it would be good to make this clear at the end of the paper.

*Colour scales.* Often diverging colour scales are used for sequential data (for example Figure 3a and 3b). I encourage the authors to change the colour scales on these plots to sequential colour scales as this is a more appropriate colour scale for sequential data values. There is a nice discussion of this issue in Section 3.2 of this preprint for HESS by Michael Stoelzle and Lina Stein: https://hess.copernicus.org/preprints/hess-2021-118/

**Minor/Technical corrections**

L20 and L57 'data basis' – unsure what is meant here. Can it be rewritten so it is clear?

L42 'are probably known to a broader audience' – can you make a more pertinent point here? What is significant about these particular missions?

L68 'and the United Kingdom' – this should be 'and Great Britain'

L89 'in nine different countries' – I would remove the word 'different' here.

Figure 1. Can you add the size of the circle to the legend in the plot making it clear how the circle size relates to catchment area?

Table 1. I don't think Table 1 adds much to the paper and would move to supplementary information.

L159 'respectively 61 runoff time series' – this doesn't make sense.

L165 It is not just changes in channel profile that lead to incorrect runoff calculation but also extrapolation of the rating curve, backwater effects etc. It may be worth expanding this a little and citing McMillan et al (2012) here (https://doi.org/10.1002/hyp.9384).

L171. When you aggregated the hourly timeseries to daily – what time period do you use?  For example, daily flow timeseries in the UK is the mean river flow in a water-day, (09.00 to 08.59 GMT, for example; 09.00 1st December to 08.59 2nd December).

L184. Personally I would not interpolate any timeseries and leave this up to the data user.

Fig 3. Figure 3a legend title should be 'Start of continuous data record'. I also think the x-axis on the histogram in Figure 3b is incorrect – should go from 0 – 100%?

L190. I am unclear what you mean by 'gauge hierarchies'?

L270. 'for each of the 3 different basin delineations' – you don't need 'different' here and can be worded as 'for each of the 3 basin delineations'. Also L423, L601 and L602 – you don't need the word 'different' here.

L310. What do you mean by 'large notches in catchment shape'?

L555. What do you mean by 'herding'?

L605. I disagree that 'These uncertainties have been addressed'. They have been considered and discussed but I wouldn't say they have been addressed as many are not quantified in the dataset.

---

## Author Comment (AC2)

**Reply to the comments of Reviewer 1 (https://doi.org/10.5194/essd-2021-72-RC1) on the manuscript "LamaH | Large-Sample Data for Hydrology and Environmental Sciences for Central Europe" submitted to ESSD (https://doi.org/10.5194/essd-2021-72).**

Dear Mathis Messager,

we pay you our gratitude for your very thorough review and the valuable inputs. We have rarely seen such a comprehensive review before. Your evaluations with R (inspection files) were also very useful and helped to improve the dataset. Please find our answers to your comments below. Your comments are displayed in italics and colored with a blue font. Our reply is written with black font, while text lines from our submitted manuscript are colored in orange-brown font (proposed text passages to delete are additionally crossed out) and suggested new text passages in green font.

Best,

Christoph in behalf of all authors

**A) COMMENTS IN MANUSCRIPT**

**1.1** *Similarly to the CAMELS datasets, consider using the dataset name "LamaH-CT" to allow for future iterations of this model in other regions.*

Thank you for the suggestion. The first CAMELS dataset also has no geographic extension, but we propose to change the title to "LamaH-CE" (for LamaH - Central Europe). In the manuscript, we would keep the term as "LamaH".

**1.2** *I suggest removing "also"*

Thanks for the comment, we have checked the manuscript and deleted unnecessary uses of the term "also".

**1.3** *Please go through manuscript and remove extraneous instances of the word "also" which is found throughout. In some instances, it would be clearer/more correct to replace with "and".*

Please see 1.2)

**1.4** *To optimize the inclusion of river network topology in applications, recommend associating the gauges with the networks so that routing may be done both by catchment and by river segment.*

Thank you for that valuable suggestion. We propose to add two new attributes (nrs_euhyd, nrs_rivat), which indicate the ID of the gauges next river segment from the stream networks "Eu-Hydro – River Network Database" and "RiverATLAS".

Table A1.

| nrs_euhyd | "OBJECT_ID" of the next river segment from stream network "Eu-Hydro – River Network Database" [f] | - | EU-Hydro - River Network Database (EEA, 2019) |
|---|---|---|---|

| nrs_rivat | "HYRIV_ID" of the next river segment from stream network "RiverATLAS" [f] | - | HydroATLAS (Linke et al., 2019) |

[f] End of the river segment (to which the attributes within the river network refer) can sometimes be rather far from the gauge. If a single river segment extended over several gauges, the ID of the river segment was only indicated at the most downstream gauge.

**1.5** *Lena river*

Thank you, we will change it according to your suggestion.

**2.1** *"their"*

Agreed, we will change it according to your suggestion.

**2.2** *"including"*

Thank you for the suggestion, we will change it according to your suggestion.

**2.3** *"large-sample hydrologic applications"*

Agree, we will change it according to your suggestion.

**3.1** *It seems that the first sentence is meant as the beginning of a comparison with other datasets, but there is no mention of basin size in other CAMELS datasets. In addition, the statement that focusing only on headwater basins (which in my understanding tends to be smaller) does not logically flow with the following sentence that questions the validity of aggregating meteorological and catchment properties over thousands of kilometers. I agree with the authors that providing multiple types of basin delineations is really valuable. However, it seems that this paragraph is not fully consistent in its argument.*
*1. The U.S. dataset has a median basin size of 330 km2 (an order of magnitude less than "thousands of square kilometers").*
*2. I am not familiar with the term "observed" basins. however, headwater to me would imply that these are usually small basins close to the headwaters. If the authors meant the entire drainage area upstream of streamflow gauging stations rather than only the local catchment, then this should be clarified in similar language as the next paragraph.*

We fully agree that this paragraph was not consistent and mixed up several issues. We propose to change the paragraph according to the following:

~~Here, the basins are mostly small- to medium-sized (median basin size of 330 km², with a range of 4 km² to 25 800 km²). Further CAMELS datasets for Chile (Alvarez Garreton et al., 2018; 516 catchments), Brazil (Chagas et al., 2020; 897 catchments), and the United Kingdom (Coxon et al., 2020; 671 catchments) followed later. CAMELS datasets always represent a composite of hydrometeorological time series and static catchment attributes aggregated to observed (headwater) basins. The~~

Further CAMELS datasets for Chile (Alvarez-Garreton et al., 2018; 516 catchments), Brazil (Chagas et al., 2020; 897 catchments), and Great Britain (Coxon et al., 2020; 671 catchments) followed later. CAMELS datasets always represent a composite of hydrometeorological time series and static catchment attributes aggregated to polygons, which cover the full upstream area. The question, how reasonable and applicable meteorological and catchment attributes are, when aggregated to the full upstream area, is critical, especially for large basins.

We have removed the term "headwater" from the manuscript and replaced it mostly with the term " independent catchments, covering the full upstream area". The term "observed" was replaced in the manuscript by "gauged". We hope that our rewording leads to a better understanding.

**3.2** *also*

Please see 1.2)

**3.3** *Similar note: I am not familiar with the term "orographic catchment area". Maybe a matter of sub-fields in hydrology. Please fully define what is meant to be from what I understand "the entire upstream drainage area"?*

Thank you for your input. The term "orographic" is quite common in German and means that the basin is delineated only based on (visible) terrain properties. We propose to replace "orographic" by the term "topographic" and suggest to add a short explanation to the manuscript:

…topographic (delineated only considering terrain features and ignoring potential subsurface cross-basin flows)

**3.4** *Please adjust coordinates to have consistent precision*

Agree, we set the number of decimals to 1.

Fig. 1, 8.2°E / 48.1°N

**4.1** *To shorten article and limit confusion with the various types of delineation, I suggest removing Table 1 and the focus on the regions (1-11), as these are not included in the dataset. This would also make Fig. 1 more readable. In addition, Fig. 1 would be most useful if a given background color was assigned to the Danube's catchment, and another color given to catchments outside the Danube's (and including in the legend the meaning of the three colors).*

Thank you on that. We propose to move Table 1 to Appendix B and to change the background color in Fig. 1 according to the larger river-regions (Danube A, Danube B, Rhine and Elbe). Water from river regions declared with "Danube B" joins the Danube outside Austria. According to an input from Reviewer 2, we have also added a legend in Fig. 1 showing the size of the circle in relation to the catchment area. We would like to keep the numbers of the river regions in the plot, as we refer to them in further text passages (e.g. in chapter 5.8).

[Figure]

Fig. 1

**4.2** *This could be removed from this section and transferred to Chapter 3 as the reader does not know what basin delineation A is yet.*

Thank you for the comment. We propose to remove the number of gauges "882" from the manuscript and move the section to the beginning of chapter 4.1. There the basin delineations and the number of gauges are described/mentioned.

**4.3** *"network"*

Agree, we will change it according to your suggestion.

**5.1** *Is this intended to convey "Most meteorological time series and catchment properties included in LamaH are based on global datasets"? If so, I suggest writing it as such. otherwise, this sentence reads as a general statement about global datasets.*

Thank you for your comment. We propose to reword the sentence as follows:

Most meteorological time series and catchment attributes included in LamaH are based on global datasets, which were provided either in raster or vector form.

**5.2** *were (otherwise, it implies that they are provided in Lamah)*

Please see 5.1)

**5.3** *"computed/calculated across"*

Thank you, we will change it according to your suggestion.

**5.4** *This sentence is not grammatically correct, please adjust.*

*In terms of content.*

*This process is unclear to me as described here. Could you please explain more how this step was conducted as it is crucial to the validity of the data. e.g. what HydroATLAS level was used? In which cases was the HAO used vs. the HydroATLAS? It seems that using a single dataset would have led to more consistency. I recommend using a single source of catchment geometric data. If the authors want to also include a more precise source of catchments, but this source is only available for a subset of the study domain and provided in addition to the main dataset. In addition, the optimal way to delineate catchments would not be to use existing catchment delineations which may not match the exact position of the gauges but rather to use a hydrologically conditioned DEM or an existing flow direction grid (e.g. https://www.hydrosheds.org/page/availability).*

We fully agree that the description of deriving the polygons, representing the topographic catchments, may lead to misunderstandings. We thank you for your comment and propose to change the paragraph as follows:

~~Consequently, aggregation of spatially distributed information of the corresponding source datasets was required. As aggregation areas are catchment boundaries (Fig. 2) from the Digital Hydrological Atlas of Austria (HAO, 2007) and from HydroATLAS (Linke et al., 2019) used. The sub-basins of both data sources were combined and, if necessary, adjusted to reflect the gauges catchment area at their downstream end. In a next step, all (smaller) subcatchments, which belong to the respective upstream catchment area of an individual gauge, were combined. Each gauge is therefore assigned an aggregation area representing the overall orographic catchment area.~~

Starting point for creating the aggregation polygons (catchments) were sub-basins from the datasets Digital Hydrological Atlas of Austria (HAO, 2007; full expansion) and HydroATLAS (Linke et al., 2019; level 12), which was used for areas not covered by HAO. The sub-basin outlets of HAO agree with the gauge locations. In contrast, the catchment boundaries of HydroATLAS were partially manually adjusted to guarantee that the basin outlets of the polygons agree with the gauging station locations. Since the sub-basin delineation in HAO and HydroATLAS were aggregated to represent the complete topographic catchment area upstream of a gauge, the different resolutions in the data sets did not matter.

**5.5** *Small comment: but for consistency, please always refer to this procedure as "basin delineation A" throughout. at the moment, both "catchment delineation" and "basin delineation" are used in the text.*

Thank you, we checked the manuscript and changed the term to "basin delineation".

**5.6** *Good and important inclusion. Please report summary statistics for this ratio.*

We propose to add the following sentences to the manuscript.

Plausibility of this type of basin delineation was checked by calculating the ratio between the area of the aggregated basins and the officially, e.g. in the metadata of the gauges, declared catchment area ("area_ratio" in Table A1). The range of "area_ratio" lies between 0.89 and 1.34, with a standard deviation of 0.026. Catchments with larger deviation in area were manually checked and corrected if there was an obvious error.

**5.7** *This value is 4.0 when rounding, right?*

Right, we changed the value to "4".

**5.9** *131 247*

Agree, we will change it according to your suggestion.

**5.10** *"identical to the delineation used in the CAMELS dataset"?*

We will change it according to your suggestion.

**5.11** *I am unclear about the meaning of this sentence. Please clarify.*

We meant that local features are reflected several times (but with a different weighting) in the data set, if catchment areas of gauges partially overlap (e.g. the catchment areas of 2 gauges, one downstream of the other, and a local feature which is covered by both catchments). We suggest to delete this sentence.

**5.12** *I recommend breaking up this paragraph in 2 around here to improve visual structuring of section.*

Thanks, we will break up the paragraph according to your suggestion.

**6.1** *I really appreciate the mention of the precise attribute names in the main text! This makes for consistency between the manuscript and the dataset. It is then possible to quickly search for the term in the manuscript while working with the dataset. If not already the case, I recommend doing it throughout.*

Thank you for your comment, we have ensured consistency in the manuscript.

**6.2** *"for" or "resulting from"*

Agree, we will change it according to your suggestion.

**6.3** *Maybe start a new paragraph here for clarity.*

Thank you, we will break up the paragraph according to your suggestion.

**6.4** *"referred to as"*

Agree, we will change it according to your suggestion.

**6.5** *I appreciate the detailed walkthrough of the figure content. This is a very useful figure to clarify the delineation methods. To maximize readability of this figure, I encourage:*

*- Removing the background or making it much lighter. The hillshade is very nicely done, but it adds a lot of visual noise to the figure at the moment.*

*- include a very short description for each basin delineation type: e.g. "total upstream drainage area", "immediate catchment", "reference catchment".*

*- Add graphical legend. While useful, the caption is very long and it requires careful attention to extract the meaning of the various cartographic elements. A graphical legend would help understand the figure more quickly/easily.*

*- Although obvious, there is no plot lettering (e.g. "plot b and c"). I defer to editor on that point.*

Thank you very much for your comments on Fig. 2. We propose to change the headers of the plots, increase the brightness of the background, add a legend in plot c) and shorten the caption of Fig.2:

[Figure]

[Figure]

[Figure]

**Fig. 2: Types of basin delineations in LamaH shown with an example.**  Plot a) Basin delineation A (similar to the well-known CAMELS datasets): The aggregation area corresponds to the topographic catchment area of a gauge. In plot a), the aggregation area of gauges 56 and 57 overlaps with that of gauge 58, and the aggregation area of gauge 55 overlaps with that of gauges 56 and 58 (indicated by the different color tones). Plot b) Basin delineation B: The aggregation areas in this method considers the difference area (intermediate catchments) between the topographic catchment area of the respective gauge and the catchment area of the next upstream gauges. Consequently, there are no overlaps, but a gauge hierarchy is necessary. The hierarchy of the gauges 54, 55 and

57 is 1, because there is no upstream gauge. Gauge 56 has the hierarchy 2, because gauge 55 with hierarchy 1 is upstream. Hierarchy 3 is assigned to gauge 58, because there is at least one gauge with hierarchy 2 (gauge 56) in the upstream area. Plot c) Basin delineation C: Similar to basin delineation B, but only uninfluenced or low-influenced gauges / catchments (see chapter 5.8) are considered. In plot c), it is assumed that gauges 54 and 56 are strongly influenced. Consequently, these two gauges are excluded from the basin delineation. The aggregation area of gauge 58 (now hierarchy 2) includes the intermediate catchment area of gauge 56. Source of background satellite image: Google © 2020 TerraMetrics, Kartendaten © 2020. Source of stream network: TYROL (2020).

**7.1** *There are 3 pairs of gauges with duplicated coordinates. Maybe merge if they are associated with overlapping time series or delete if truly duplicates? if not, please clarify somewhere.*

The level of detail of your review is really impressive. It is correct, that there are 3 gauges with identical coordinates. These "duplicates" are attributed with "degimpact = x", because they, for instance, do not considering a mill channel (while the other with the same coordinates does). Please see also 24.2), where we have added the information regarding duplicates in the row "x" of Table 2.

**7.2** *"125 and 61 runoff time series, respectively"*

Thank you for your comment. We propose a rewording of the sentence:

The hydrographical services of the German federal states Bavaria (GKD, 2020) and Baden-Württemberg (LUBW, 2020) provided 125 and 61 runoff time series, respectively.

**7.3** *Please start new paragraph here for clarity*

Thank you, we will break up the paragraph following your suggestion.

**7.4** *The metadata for the gauges time series dataset Hydro_indices_beg_2017.csv is said to include all records since the beginning of recording for the station. However, as noted here, the time series were truncated to start in 1981. Either the metadata needs to be adjusted (e.g. Hydro_indices_1981_2017.csv) or the full time series (since the real beginning) should be provided.*

Thank you for the comment. We agree that the name of the file can lead to misunderstandings. By "begin" we mean the first 01 October since 1981. We propose to change the filename according to your suggestion to "Hydro_indices_1981_2017.csv" and to change the passages in the manuscripts from "" to "first 1 October after 1981".

**7.5** *"including"*

We propose to change the term to "even" and add the name of the corresponding attribute:

Some time series included gaps, even after checking by the hydrological services ("gaps_pre" in Table A1).

**7.6** *Does this imply that gaps > 6 h were completely left unchanged, or that the first 6h of the gap was imputed? I assume the former, but this is not fully clear in this sentence. I suggest rephrasing as "gaps of up to 6h were filled with linear interpolation during our processing. Any remaining gaps (> 6h) in ..."*

Thank you for your comment. Yes, the former is correct, gaps > 6 h are completely left unchanged. We propose to reword the sentence according your suggestion:

Gaps of up to 6 hours were filled with linear interpolation during our processing, if the number of consecutive gaps was less than 7. Any remaining gaps (> 6h) were marked with the number -999.

**7.7** *I recommend including both a gap_pre and gap_post column for users to easily assess the degree of interpolation without having to summarize the gap datasets themselves.*

Thanks for the suggestion, we have added the attribute "gaps_pre" and renamed the existing attribute from "gaps" to "gaps_post". Furthermore, we have adjusted the terms in the manuscript.

Table A1.

| gaps_pre | Fraction of gaps in the raw hourly runoff time series | ‰ | see above |
|---|---|---|---|
| gaps_post | Fraction of gaps in the hourly runoff time series after our processing (linear interpolation, up to 6h) | ‰ | see above |

**8.1** *Great!!*

Thank you!

**8.2** *"Beginning of continuous data recording"*

We propose to change the term to "Start of continuous data recording".

**8.3** *I struggled to grasp the meaning of this sentence. Please clarify, probably splitting into two sentences.*

Thank you for the comment. We propose to reformulate the sentence as follows, as the upscaling approaches are moved to chapter 3) according to your suggestion (see also 11.1).

The aggregation was done by calculating the area-weighted arithmetic mean (upscaling approach 1).

**8.4** *"statistically summarized/aggregated"?*

Agree, we will change it according to your suggestion.

**9.1** *I recommend making it explicit how that this was computed from the gauging stations records, I believe (e.g. "Q, as recorded at the gauging station").*

Thank you, we propose to add the sub-clause according your suggestion.

The difference between long-term mean precipitation (P) and runoff height (Q), as recorded at the gauging station, should be equal to the total evapotranspiration (ETA) in a fulfilled water balance.

**9.2** *I assume that runoff height was computed using the "area_calc" right? Please confirm. For a few stations with area_ratio most different from 1, the difference in actual vs. calculated catchment may partly explain the divergence.*

Yes, the runoff height was calculated using "area_calc". We believe that the area of the aggregated polygons describes the topographic catchment area more consistently than the officially declared. The official stated catchment "area_gov" area may be obtained differently in the individual countries / federal states (e.g. considering the hydrographic catchment area). Furthermore, we checked the catchments with a low/high "area_ratio" manually and can therefore ensure a sufficient mapping of the topographic catchment area.

**9.3** *Good job in including a comparison.*

Thank you!

**9.4** *"between"*

Agreed, we will change it according to your suggestion.

**9.5** *"and" (otherwise, this may seem like another ratio)*

Thanks, we will change it according to your suggestion.

**10.1** *These sentences are grammatically a bit off. Please correct.*

Thank you for the comment. We propose to reword the sentence as follows:

**In a), b) and c), values are only plotted for basins with observations for the period 01.10.1989 to 30.09.2009 (717 basins). Further, in a) and c), values are only plotted for basins not affected by artificial water input or withdrawal, karstic springs or high infiltration (594 basins; see chapter 5.8).**

**10.2** *Please add the number of gauges/basins for both statements.*

Please see 10.1

**10.3** *Should this be 859? It was my understanding that basins could not be delineated for 23 gauges.*

Thank you very much for the comment. It should be of course "859".

**11.1** *Because this explanation is applicable to ERA-Land data processing and the explanation in the ERA-Land section is a bit confusing, this section (catchment attributes) could be placed before the meteorological attributes section so that the second approach can be referred to directly when talking about ERA-Land? To clarify explanation, I recommend reversing the order of explanation for the approaches. This would help clarify exceptions in upscaling approach 1: base on my understanding, what is now called upscaling approach 1 could simply say "in the case of small catchments where no or only one raster centroid intersects the basin delineation. The upscaling approach 2 was used."*

Thank you for your valuable suggestion. We propose to move this section to chapter 3) and to reword / re-index it according to your suggestion. Due to the re-indexation, we have also adjusted the notes of the tables in the appendix.

~~Furthermore, the upscaling is performed by two different approaches: 1) In the first approach (referred as "upscaling approach: 1") the aggregation is based on all the raster cells, which centroids are located inside the catchment, whether the catchment completely covers them or not. Especially at small catchments it is possible that no or only one raster centroid intersects the basin delineation. In this case the aggregation is calculated from all contributing raster cells using an area-weighted mean. Upscaling approach 1 is mainly used for relatively fine-gridded data sources (< 1 km), since it is not that computing-intensive and potential inaccuracies are negligible. 2) The second approach of upscaling (referred as "upscaling approach: 2") exclusively performs the aggregation area-weighted. Upscaling approach 2 is used for coarser gridded and vectorial data sources. The applied approach is indicated in the corresponding tables in the appendix.~~

**Aggregation of the spatially distributed information of the used basic datasets for meteorological time series and various static attributes is performed for each of the 3 basin delineations by calculating the area-weighted arithmetic mean (otherwise indicated in the text). This method of aggregation is used for coarser gridded and vectorial data sources and is referred in the following as "upscaling approach 1". The alternative "upscaling approach 2" is based on all the raster cells, which centroids are located inside the polygon ("aggregated basins" in Fig. 2: Types of basin delineations in LamaH shown with an example.  Plot a) Basin delineation A (similar to the well-known CAMELS datasets): The aggregation area corresponds to the topographic catchment area of a gauge. In plot a), the aggregation area of gauges 56 and 57 overlaps with that of gauge 58, and the aggregation area of gauge 55 overlaps with that of gauges 56 and 58 (indicated by the different color tones). Plot b) Basin delineation B: The aggregation areas in this method considers the difference area (intermediate catchments) between the topographic catchment area of the respective gauge and the catchment area of the next upstream gauges. Consequently, there are no overlaps, but a gauge hierarchy is necessary. The hierarchy of the gauges 54, 55 and 57 is 1, because there is no upstream gauge. Gauge 56 has the hierarchy 2, because gauge 55 with hierarchy 1 is upstream. Hierarchy 3 is assigned to gauge 58, because there is at least one gauge with hierarchy 2 (gauge 56) in the upstream area. Plot c) Basin delineation C: Similar to basin delineation B, but only uninfluenced or low-influenced gauges / catchments (see chapter 5.8) are considered. In plot c), it is assumed that gauges 54 and 56 are strongly influenced. Consequently, these two gauges are excluded from the basin delineation. The aggregation area of gauge 58 (now hierarchy 2) includes the intermediate catchment area of gauge 56. Source of background satellite image: Google © 2020 TerraMetrics, Kartendaten © 2020. Source of stream network: TYROL (2020).**

). In case of small catchments, where no or only one raster centroid intersects the polygon, upscaling approach 1 was used. Upscaling approach 2 is mainly used for relatively fine-gridded data sources (< 1 km grid size), since it is not that computing-intensive and potential inaccuracies are negligible. The applied approach is indicated in the relevant tables in appendix A.

**11.2** *This feels redundant with section explaining the delineation techniques. i suggest removing for the sake of brevity.*
Thank you for the comment. We propose to delete the paragraph according to your suggestion:

~~Basin delineation A shows that about 34% of all 859 basins (aggregation areas) are smaller than 100 km2, 50% are between 100 and 1 000 km2, 14% are between 1 000 and 10 000 km2, and about 2.8% are larger than 10 000 km2. Large catchment areas are especially present for the gauges at the Danube and its larger tributaries (Fig. 5a). One reason for using multiple basin delineations is the reduction of aggregation areas and thus providing a more representative representation of local conditions and maintain natural variability. When applying basin delineation B, about 45% of all 859 aggregation areas have an area of less than 100 km2, 52% between 100 and 1 000 km2 and only 2.3% have an area above 1 000 km2.~~

**11.3** *"vertical" error (because SRTM products include horizontal error as well). It is worth mentioning that the usual vertical error is 5-9 m.*
Agree, we have added "vertical".

SRTM features a grid size of 30 m and provides a maximum global absolute vertical error of 16 m at a 90% confidence interval, while accuracy decreases with increasing elevation and slope (Farr et al., 2007).

**11.4** *Example of descriptive section which could be removed for brevity.*
We internally discussed deleting the mentioned section, where the spatial distribution of the catchment attributes is described (according to your suggestions). We understand the merit of shortening the manuscript. Deleting the section would also mean that the plots would become obsolete since they need some short description. Deleting would also mean that the overview regarding the spatial distribution of the catchment properties is lost. In other CAMELS papers there is however frequently a clear subdivision between "data and methods" and "spatial variability". However, such a division would lead to a lengthening of the article, since the description/discussion of the spatial distribution would then have to be provided for all catchment attributes shown in the plots (in order to be consistent). Accordingly, we would prefer to keep the structure of the manuscript. Therefore, we hope for your understanding.

**12.1** *the strm_dens attribute in table A3 mentions HydroATLAS and the Australian datasets as sources, which would be problematic (see my comment on table A3). Please clarify whether only the EEA dataset was used.*
Thank you for the indication. We referred to the two datasets "HAO" and "HydroATLAS" in addition to "EU-Hydro - River Network Database" in Table A3, because the catchment area was derived from them. We of course only used "EU-Hydro -

River Network Database" as stream network for calculating the stream density. We propose to delete "HAO, (2007)" and "HydroATLAS (Linke et al., 2019)" from Table A3 as this can lead to confusion:

Table A3.

| strm_dens | Stream density $D_F$, ratio of lengths of streams $L_F$ to the catchment area A (area_calc), $D_F = \frac{\sum L_F}{A}$ | km km$^{-2}$ | EU-Hydro - River Network Database (EEA, 2019), HAO (2007), HydroATLAS (Linke et al., 2019) |
| --- | --- | --- | --- |
| | | | |

**12.2** *I recommend writing "elongation ratio" instead of length extension for consistency.*

Agree, we will change it according to your suggestion.

**14.1** *"[...] only computed for those gauges which cover [...]", the comma otherwise makes the sentence a bit off grammatically.*

Thank you, we will change it according to your suggestion.

**14.2** *Two comments on this:*

*1. From my understanding of the dataset, the full hydrological indices are provided for only 859 gauges. Why not 882? The back and forth between the 859 and 882 is slightly confusing at times. I recommend to either provide these hydrological indices or to remove the remaining 23 gauges from the manuscript altogether?*

Thank you for your comment. The hydrological indices were not computed for those 23 gauges, because there is no defined area to compute the runoff heights. Although we understand that sometimes the back and forth is confusing, we would like to keep the 23 gauges in the dataset because they might provide valuable information (e.g. runoff in artificial channels or karst springs). In response to your comment, we tried to improve clarity by restructuring the text (see 4.2 of section A).

*2. 13 of the provided 859 records have NAs in Hydro_indices_beg_2017.csv. I am not sure why. From my inspection, it seems that the 13 excluded ones are either short or have a full year missing? Can you clarify please?*

Hydrological signatures are not computed if a) fraction of gaps in the investigation period is more than 5%, and b) less than 5 full hydrological years are recorded. 13 of the provided 859 records could not meet these criteria. But thank you for the hint, and we propose to reformulate the section at the beginning of chapter 5.3 as follows:

The indices were computed for those gauges, which cover the whole period of investigation (717 gauges). However, the evaluations for the entire period of record (first 01. October to 30. September 2017) are additionally made available within the dataset. However, evaluations for the entire period of record (first 1 October after 1981 to 30 September 2017) are additionally made available if at least 5 full hydrological years are recorded. 4 gauges do not meet this requirement. Hydrological signatures are calculated, only if the fraction of gaps is less than 5% for both evaluation periods.

In addition, we suggest to add the criteria a) to the Tables A5 (Hydrological signatures) and the criteria a) and b) to Table A10 (Attributes in the accompanying file "Water_balance.csv") as a footnote.

*In addition, ID 675 is NA in indices_beg_2017.csv but not in indices_1989_2009.csv*

The fraction of gaps is 7.3% (NA, because >5%) in the period 01 Oct 1981 to 30 Sept 2017, while 2.5% (values, because <5%) in the period 01 Oct 1989 to 30 Sept 2009.

**16.1** *Please re-arrange sentence for clarity (see comment on similar point above).*

Thank you, we propose to change the sentence as follows:

**Only gauges are plotted, which cover the period 1 October 1989 to 30 September 2009.**

**17.1** *Example of section that could be removed (or moved elsewhere) for brevity.*

Please see 11.4)

**21.1** *"of...with" or "between...and"*

We propose to use "between and", thank you.

Further interrelationships between the various grain size fractions and the dominating bedrock are recognizable:

**24.1** *Impressive and very appreciated addition.*

Thank you very much! We really appreciate your feedback!

**24.2** *For clarity, I highly recommend synthesizing this information in a table, as I struggled following it. A useful complementary format for this could be a graphical decision tree.*

We agree that it is hard to follow all the statements regarding the allocation of the degree of impact and propose to add a table instead:

[revised manuscript text omitted]

**25.2** *If a good quality dataset of reservoirs is available, I suggest including additional statistics regarding the degree of influence by lakes and reservoirs. Attributes could include surface area and volume of upstream lakes and reservoirs (separately), and a Degree of Regulation measure (DOR, see Lehner B et al 2011 High-resolution mapping of the world's reservoirs and dams for sustainable river-flow management Front. Ecol. Environ. 9 494–502).*

Thank you very much for the valuable suggestion. For water reservoirs, we propose to add the corresponding "GRAND_ID" from the GRanD dataset (Lehner et al., 2011) and the "DAM_ID" from the GOODD dataset (Mulligan et al., 2020) in a novel shapefile (Impacts.shp). This shapefile allocates the (anthropogenic) impacts on the runoff process and measurement based on basin delineation B. However, we do not intend to derive further statistics because only a few water reservoirs in LamaH are covered by the GRanD and the GOODD dataset. Users can however easily derive dam statistics with the linking-attributes "GRAND_ID" and "DAM_ID" on their own, if necessary. We propose to add a new text section, which describes the novel shapefile (Impacts.shp) and the linking IDs to the two dam datasets.

Geo-localization of the impacts is provided by the shapefile "Impacts.shp", which includes links to the dam datasets GRanD ("GRAND_ID"; Lehner et al., 2011) and GOOD ("DAM_ID"; Mulligan et al., 2020) to ensure fast access those attributes.


We also created a text-file reflecting the cross-basin water transfers ("typimpact" = E or I) and propose to add a table in the Appendix which shows the attributes of that novel file:

**Table A11: Attributes in the accompanying file " Crossbasin_water_transfer.csv" ᵃ**

| Attribute | Description | Unit |
|---|---|---|
| ID | ID of sourcing basin | - |
| To_ID | ID of receiving basin | - |
| cross_region | Flag if river region (**Fehler! Verweisquelle konnte nicht gefunden werden.**, Table B1) is crossed (0 - no, 1 - yes) | - |
| lower_thres | Lower threshold for water cross-basin transfer | $m^3\ s^{-1}$ |
| upper_thres | Upper threshold for water cross-basin transfer | $m^3\ s^{-1}$ |
| estimated | Flag if "lower_thres" and "lower_thres" are estimated (1) due to lack of publicly available information, otherwise this information is taken from governmental registers, system specifications, etc. (0) | - |
| model | Flag if the cross-basin transfer is implemented in the hydrological model COSERO (0 - no, 1 - yes); COSERO can only transfer water from a source basin to one single receiving basin | - |

ᵃ Derived for basin delineation B.

**27.1** *I suggest refraining from using the term headwater catchment for this purpose (I now understand how it is being used in this manuscript), as this term is often also used to designate small catchments close to the headwaters (rather than the full drainage area upstream of a given pourpoint as it is used here).*

We have removed the term "headwater" from the manuscript. Please see 3.1)

**27.2** *Prior to the publication of this manuscript, please include a code availability statement and provide the code used in generating the dataset and figures.*

Thank you for the comment. We agree to add relevant codes to reproduce the dataset. We propose to add the following paragraph:

**Code availability.** We have used R-Codes from Nans Addor (Addor, 2017b) for reproducing the climatic (Table A4) and hydrological (Table A5) indices as well as for creating the Figures 3, 5 to 13a/c. The color schemes in the plots are often based on ColorBrewer 2.0 (Brewer, 2021). Further relevant R- and Python-scripts for reproducing the dataset are available in the folder "G_appendix".

**37.1** *Please add attribute reference table for water_balance.csv*

We propose to add Table A10 in the appendix. This table also contains data for long-term PET of ERA5-Land (Evidence for argumentation in chapter 4.2) as well as precipitation sums of the alternative datasets ERA5 (Hersbach et al., 2020), CHRIPS v2 (Funk et al., 2015), and MSWEP v2.2 (Beck et al., 2017; 2019) displayed in Fig. 4.

**Table A10: Attributes in the accompanying file "Water_balance.csv" [e]**

| Attribute | Description | Unit | Data source |
|-----------|-------------|------|-------------|
| P | Mean precipitation of ERA5-Land [a,b] | mm d$^{-1}$ | ERA5L (Muñoz Sabater, 2019a) |
| P_CRPS | Mean precipitation of CHIPRS Daily v2.0 [a,b] | mm d$^{-1}$ | CHIRPS Daily v2.0 (Funk et at., 2015) |
| P_ERA5 | Mean precipitation of ERA5 [a,b] | mm d$^{-1}$ | ERA5 (Hersbach et al., 2020) |
| P_MSW | Mean precipitation of MSWEP v2.2 [a,b] | mm d$^{-1}$ | MSWEP v2.2 (Beck et al., 2017; 2019) |
| PET | Mean PET of ERA5-Land [a,b] | mm d$^{-1}$ | ERA5L (Muñoz Sabater, 2019a) |
| ET0 | Mean ET0 of Climate Database v2 [a,c] | mm d$^{-1}$ | Global Aridity Index and Potential Evapotranspiration (ET0) Climate Database v2 (Trabucco and Zomer, 2019) |
| ETA | Mean actual evapotranspiration of ERA5-Land [a,b] | mm d$^{-1}$ | ERA5L (Muñoz Sabater, 2019a) |
| Q_1 | Mean runoff height [b,f,h] | mm d$^{-1}$ | BAFU (2020), CHMI (2020), GKD (2020), HZB (2020), LUBW (2020) |
| Q_2 | Mean runoff height [d,g,h] | mm d$^{-1}$ | see above |

[a] Computed for basin delineation A. [b] Period 1 October 1989 to 30 September 2009. [c] Period 1970 to 2000. [d] Period from the first 1 October in the time series after 1981 to 30 September 2017. [e] No values, where corresponding gauge is attributed with "degimpact" = x (**Fehler! Verweisquelle konnte nicht gefunden werden.**). [f] No values for gauges, whose time series do not cover the full period. [g] Values only if at least 5 years of observations are available. [h] No values if there are more than 5% gaps in the calculation period.

**37.2** *I highly encourage adding the ID of the river segment in the RiverATLAS and EEA stream networks which correspond to the gauges when possible. This would be a great addition that would further enable users to route through these networks, and combine the attributes from this study with RiverATLAS attributes.*

Thank you for your valuable suggestion. We have added two additional attributes, which indicate the IDs of the gauges nearest stream segments. Please see 1.4)

**38.1** *"Beginning"*

Thank you, we propose to use the term "Start".

Start of continuous (hourly) runoff data recording

**40.1** *same comment as strm_dens below*

Thank you for the comment. For the derivation of the attribute "mvert_dist" both the location of the gauges and the shape of the catchment area was needed. We propose to cite the runoff data provider instead of the sub-basin datasets:

Table A3.

| mvert_dist | Horizontal distance from the farthest point of the catchment to the belonging gauge (length axis) | km |  BAFU (2020), CHMI (2020), GKD (2020), HZB (2020), LUBW (2020) |
|---|---|---|---|

**41.1** *Because EU-Hydro, HAO, and HydroATLAS were derived at different resolutions (I believe), it is not possible to compare stream densities among them. Therefore, using different networks for different catchments would lead to inconsistencies. I recommend computing stream densities from a single network.*

Thank you for the comment. The stream density was of course computed using only one stream-network dataset. But the two sub-basin datasets (HAO and HydroATLAS) were also used for obtaining the catchment area. Please see 12.1)

**41.2** *Why wasn't the global aridity index from Trabucco and Zomer directly used? It appears that for the sake of consistency, it would be better than combining ERA5L data with their ET0.*

Thank you for your input. We propose to add an additional attribute "arid_2" and rename the existing attribute to "arid_1" instead of overwriting the attribute "arid". The corresponding text passages in the manuscript will be adjusted and Fig. 6c will show the spatial distribution of "arid_2".

Table A4.

| arid_1 | Aridity, computed as the ratio of mean ET0 [b,e] (from Climate Database v2) and mean precipitation [a,d] (from ERA5-Land) | - | ERA5L (Muñoz Sabater, 2019a), Global Aridity Index and Potential Evapotranspiration (ET0) Climate Database v2 (Trabucco and Zomer, 2019) |
|---|---|---|---|
| arid_2 | Reciprocal value of aridity index from Climate Database v2 [b,e] | - | Global Aridity Index and Potential Evapotranspiration (ET0) Climate |

| | | | | Database v2 (Trabucco and Zomer, 2019) |
|---|---|---|---|---|

**45.1** *Upon examination of the catchment attributes, the sum of texture fractions > 1. Please check that these statistics are valid.*

Thank you for the comment. The sum of the texture fractions sand/soil/clay should be 1, however not for sand/silt/clay/gravel. We propose to move the note "of soil material < 2 mm" from the bottom of the table to the attribute´s description and to add the term "of overall soil" in the description of "grav_fra" and "oc_fra" to make it more obvious.

Table A8.

| sand_fra | Sand fraction (of soil material < 2 mm) [a,b,c] | - | see above |
|---|---|---|---|
| silt_fra | Silt fraction (of soil material < 2 mm) [a,b,c] | - | see above |
| clay_fra | Clay fraction (of soil material < 2 mm) [a,b,c] | - | see above |
| grav_fra | Fraction of gravel (of overall soil) [a,b,c] | - | see above |
| oc_fra | Fraction of organic material (of overall soil) [a,b,c] | - | see above |

[a] Areas marked as water or bedrock were excluded from calculation. [b] Aggregation weighted by depth over the different soil layers. [c] Upscaling approach 2.

**B) FURTHER COMMENTS IN REVIEW REPORT** (which are not already addressed in section A)

**1.2** *If available, please add a link to the HAO reference.*

The link to the homepage is not really static, as the name of the belonging ministry frequently changes after elections… We propose to add instead the ISBN in the reference.

HAO: Hydrological Atlas of Austria (digHAO), 3. Delivery, Federal Ministry of Agriculture, Regions and Tourism – Hydrographic Central Office, Vienna, Austria, ISBN 3-85437-250-7, 2007.

**2.1** *However, it is important to note further in the manuscript that very limited data quality checking was performed by the authors for the streamflow gauging stations time series. I do not suggest that additional QA/QCing be performed, simply that this be mentioned as a disclaimer. At the moment, the only QA/QCing performed on these time series originates from the providing agencies. Gauging stations time series are notoriously prone to artefacts (for various reasons), and additional QA/QCing is often needed (e.g. Gudmundsson et al. 2019, Zimmer et al. 2020). I recommend that additional resources be provided to readers for assessing the extent and nature of this checking if possible.*

Thank you for your comment. We fully agree that additional quality flags are really valuable and propose to add flags for consecutive time steps ("qceq" in Table A1) and outliers ("qcol") to every runoff time step (derived according to chapter 2.3 in Gudmundsson et al. 2018). We think that the runoff time steps with gaps (-999) are easy to filter anyway and do not necessarily need an extra column.

Table A1.

| qceq | QC flag which is set, if runoff remains equal in at least 10 consecutive time steps (daily or hourly); chapter 2.3 in Gudmundsson et al. (2018) [e] | 0 or 1 | see above |
|------|-----|--------|-----------|
| qcol | QC flag which is set, if runoff value is classified as outlier; chapter 2.3 in Gudmundsson et al. (2018) [e] | 0 or 1 | see above |

[e] Visible in daily and hourly runoff time series.

Thank you very much, we appreciate your detailed investigation. Following your finding, we found out that there might be some issues in QGIS with integers containing too many digits (changing to character format, see https://issues.qgis.org/issues/14691). However, we checked the number of digits (should be < 10) of all shapefile-attributes and ensured integer format for the attributes "ID" and "NEXTDOWNID". The attribute "NEXTUPID" requires character format in order to list multiple IDs.

**4.3** *I do not believe that "formal" metadata (e.g. ISO 19115 and ISO 19139) are provided with the dataset, beyond an information text file.*

We have added metadata to every shapefile by a .qmd file (with QGIS, should be similar to .xml) which contains some basic information like a short description, license or coordinate system (EPSG).

---

## Author Comment (AC3)

**Reply to the comments of Reviewer 2 (https://doi.org/10.5194/essd-2021-72-RC2) on the manuscript "LamaH | Large-Sample Data for Hydrology and Environmental Sciences for Central Europe" submitted to ESSD (https://doi.org/10.5194/essd-2021-72).**

Dear Gemma Coxon,

thank you for your positive feedback as well as your valuable comments and suggestions. Please find our answers to your comments below. Your comments are displayed in italics and colored with a blue font. Our reply is written with black font, while text lines from our submitted manuscript are colored in orange-brown font (proposed text passages to delete are additionally crossed out) and suggested new text passages in green font.

Best,

Christoph in behalf of all authors

**1.** *Basin delineation and 'headwater' catchments. The delineation of the catchment boundaries is a key feature of the dataset but currently is not clear. I suggest the following:*

Thank you for your comment. We fully agree, that the delineation of catchment boundaries was not well described and the actual terminology can lead to misunderstandings.

**1.1** *Terminology needs to be much clearer. What do you mean by 'orographic catchment' (is this a commonly used term? Do you mean topographic catchment?). I don't believe you are using the term 'headwater catchment' correctly - I interpret headwater catchments as low order catchments found in the upper reaches of river basins. In which case statements like 'In contrast, however, LamaH does not only consider headwater basins' on L18 are not correct as CAMELS datasets also do not only consider headwater basins.*

We replaced "orographic" by the term "topographic" and suggest to add a short explanation to the manuscript:

(delineated only considering terrain features and ignoring potential subsurface cross-basin flows)

The term "headwater catchment" was replaced by "independent catchments, covering the full upstream area".

**1.2** *Data source of the catchment boundaries – why did you use catchment boundaries from two products? It is not clear how the catchment boundaries are combined. L114 'As aggregation areas....' This sentence doesn't make sense and needs rewriting.*

We propose to change the description of the sub-basin aggregation and hope that this makes it clearer and explains why we used two different products.

As aggregation areas are catchment boundaries (Fig. 2) from the Digital Hydrological Atlas of Austria (HAO, 2007) and from HydroATLAS (Linke et al., 2019) used. The sub-basins of both data sources were combined and, if necessary, adjusted to reflect the gauges catchment area at their downstream end. In a next step, all (smaller) sub-catchments, which belong to the

Starting point for creating the aggregation polygons (catchments) were sub-basins from the datasets Digital Hydrological Atlas of Austria (HAO, 2007; full expansion) and HydroATLAS (Linke et al., 2019; level 12), which was used for areas not covered by HAO. The sub-basin outlets of HAO agree with the gauge locations. In contrast, the catchment boundaries of HydroATLAS were partially manually adjusted to guarantee that the basin outlets of the polygons agree with the gauging station locations. Since the sub-basin delineation in HAO and HydroATLAS were aggregated to represent the complete topographic catchment area upstream of a gauge, the different resolutions in the data sets did not matter.

**1.3** *Figure 2. I find this figure a little unclear and the figure caption is currently very long. It may be worth simplifying the figure and thinking about moving some of the explanatory text in the figure caption to the main text in the paper.*

Thank you for the suggestion. We propose to add a legend to the figure and to shorten the figure caption. We also increased the brightness of the background and hope that these adjustments will make it easier to understand the plot. We hope that these adaptions allow for a good understanding of the sub-basin aggregation, the catchment delineations and gauge hierarchy.

[Figure]

[Figure]

[Figure]

Fig. 2: Types of basin delineations **in LamaH** shown with an example.  Plot a) Basin delineation A (similar to the well-known CAMELS datasets): The aggregation area corresponds to the topographic catchment area of a gauge. In plot a), the aggregation area of gauges 56 and 57 overlaps with that of gauge 58, and the aggregation area of gauge 55 overlaps with that of gauges 56 and 58 (indicated by the different color tones). Plot b) Basin delineation B: The aggregation areas in this method considers the difference area (intermediate catchments) between the topographic catchment area of the respective gauge and the catchment area of the next upstream gauges. Consequently, there are no overlaps, but a gauge hierarchy is necessary. The hierarchy of the gauges 54, 55 and 57 is 1, because there is no upstream gauge. Gauge 56 has the hierarchy 2, because gauge 55 with hierarchy 1 is upstream. Hierarchy 3 is assigned to gauge 58, because there is at least one gauge with hierarchy 2 (gauge 56) in the upstream area. Plot c) Basin delineation C: Similar to basin delineation B, but only uninfluenced or low-influenced gauges / catchments (see chapter 5.8) are considered. In plot c), it is assumed that gauges 54 and 56 are strongly influenced. Consequently, these two gauges are excluded from the basin

**2.** *Potential evapotranspiration. The analysis in Section 4.2 is really interesting and an important addition to the paper. I understand the authors decision to not include the PET data of ERA-5, but given the importance of this variable as forcing data for a large amount of hydrological models (particularly conceptual lumped hydrological models), it seems a shame not to include it as a variable.*

Thank you for the appreciation of Section 4.2 and understanding of our decision not to include the ERA-5L PET data. We fully agree with your thoughts regarding the importance of PET as input for hydrological modelling. We of course discussed this deficit of LamaH internally when preparing the manuscript, but did not find an adequate alternative data product of PET to use. For the revised manuscript / updated LamaH-dataset we propose to provide daily time series of PET estimates (Thornthwaite's approach) from the hydrological model simulations, which will be included in LamaH (see our response to the comment CC1 of Daniel Klotz in the discussion).

*It is not entirely clear how you would derive PET from the reference evapotranspiration provided. Were other global PET products considered?*

It is not intended to derive PET from ET0 (available as static attribute calculated for the period 1970-2000). We provide therefore an attribute for long-term ET0 as a substitute for PET and propose to clarify this in the manuscript:

Potential evapotranspiration (PET) can be derived from ET0 using correction factors for vegetation and soil properties (Allen et al., 1998; Hargreaves, 1994), but was not realized in LamaH.

**3.** *Code availability. Reproducibility for these large-sample datasets is key. The authors should consider making their code available alongside the dataset. I would also recommend a code availability statement to make clear the code that was used in the paper. For example, I believe you used Nans' code to calculate the hydrologic and climatic catchment attributes and it would be good to make this clear at the end of the paper.*

Thank you for your suggestion. We have already cited the codes of Nans Addor (Addor, 2017b), but can of course also explicitly mention this in the end of the manuscript. Furthermore, we will add all relevant R and Python scripts for reproducing the dataset into the "G_appendix" folder of the dataset.

**Code availability.** We have used R-Codes from Nans Addor (Addor, 2017b) for reproducing the climatic (Table A4) and hydrological (Table A5) indices as well as for creating the Figures 3, 5 to 13a/c. The color schemes in the plots are based on ColorBrewer 2.0 (Brewer, 2021). Further relevant R- and Python-scripts for reproducing the dataset are available in the folder "G_appendix".

**4.** *Colour scales. Often diverging colour scales are used for sequential data (for example Figure 3a and 3b). I encourage the authors to change the colour scales on these plots to sequential colour scales as this is a more appropriate colour scale for*

*sequential data values. There is a nice discussion of this issue in Section 3.2 of this preprint for HESS by Michael Stoelzle and Lina Stein: https://hess.copernicus.org/preprints/hess-2021-118/*

Thank you for your suggestion. We have changed the color scales from diverging to sequential for the following plots and adapted the caption of Fig 3a according your suggestion:

[Figure]

Fig. 3

[Figure]

Fig. 5

[Figure]

a) Mean daily precipitation P [mm/day]

b) Mean daily reference evapotranspiration ET0 [mm/day]

c) Aridity (ET0/P) [−]

d) Seasonality of precipitation [−]

e) Frequency of high precip. days [days/yr]

f) Mean duration of high precip. events [days]

g) Fraction of precipitation falling as snow [−]

h) Frequency of dry days [days/yr]

i) Mean duration of dry periods [days]

Fig. 6

[Figure]

a) Mean daily discharge Q [mm/d]

b) Runoff ratio (Q/P) [−]

c) Mean half−flow date [days since 01. Oct.]

d) Slope of the flow duration curve [−]

e) Baseflow index [−]

f) Discharge precipitation elasticity [−]

g) High−flow frequency [days/yr]

h) Mean high−flow duration [days]

i) Q95 [mm/d]

j) Low−flow frequency [days/yr]

k) Mean low−flow duration [days]

l) Q5 [mm/d]

Fig. 7

[Figure]

Fig. 8

[Figure]

Fig. 9

[Figure]

Fig. 10

[Figure]

Fig. 11

**5.** *L20 and L57 'data basis' – unsure what is meant here. Can it be rewritten so it is clear?*

Thanks for the comment, in L20 we propose to rewrite the sentence and in L57 to use the term the term "dataset" instead.

We describe not only the used basic datasets (e.g. for elevation) and methodology of data preparation, but also focus on possible limitations and uncertainties.

**6.** *L42 'are probably known to a broader audience' – can you make a more pertinent point here? What is significant about these particular missions?*

Sentinel, MODIS and Landsat are famous and popular remote sensing products. We propose to remove this sentence, because it doesn't really add valuable information to the manuscript.

**7.** *L68 'and the United Kingdom' – this should be 'and Great Britain'*

You are of course right - we will change it according to your suggestion.

**8.** *L89 'in nine different countries' – I would remove the word 'different' here.*

Please see 17).

**9.** *Figure 1. Can you add the size of the circle to the legend in the plot making it clear how the circle size relates to catchment area?*

Thank you for your suggestion. We have added a legend to the figure. Reviewer 1 has suggested to highlight the macro- river region (i.e. Danube A, Danube B, Rhine and Elbe). Water from river regions declared with "Danube B" join the Danube outside Austria.

[Figure]

Fig. 1

**10.** *Table 1. I don't think Table 1 adds much to the paper and would move to supplementary information.*

We agree and propose to move Table 1 to Appendix B.

**11.** *L159 'respectively 61 runoff time series' – this doesn't make sense.*

Thank you for your comment. We have reformulated the sentence.

The hydrographical services of the German federal states Bavaria (GKD, 2020) and Baden-Württemberg (LUBW, 2020) provided 125 and 61 runoff time series, respectively.

**12.** *L165 It is not just changes in channel profile that lead to incorrect runoff calculation but also extrapolation of the rating curve, backwater effects etc. It may be worth expanding this a little and citing McMillan et al (2012) here (https://doi.org/10.1002/hyp.9384).*

We fully agree with your comment. Thank you for pointing to McMillan et al. (2012). We propose to extend the sentence as follows:

Changes in channel profile, e.g. after floods with strong bedload transport, extrapolation of the rating curve or backwater effects and transient runoff conditions (runoff hysteresis) can lead to an incorrect runoff determination (McMillan et al., 2012).

**13.** *L171. When you aggregated the hourly timeseries to daily – what time period do you use? For example, daily flow timeseries in the UK is the mean river flow in a water-day, (09.00 to 08.59 GMT, for example; 09.00 1st December to 08.59 2nd December).*

Thank you for this comment and it is a really important information. In Austria (7:00 to 6:59 CET of the following day) it is quite the same than in UK. We decided to aggregate from 0:00 to 23:59, as the timestamps are probably different in each country.

However, we normally requested only the time series with hourly resolution and derived the daily time series from them. Thereby the hourly values of the respective day were used for determining the daily values (as well as for the meteorological variables), e.g. for 01.01.1981: 01.01.1981 00:00 to 01.01.1981 23:59 GMT.

**14.** *L184. Personally I would not interpolate any timeseries and leave this up to the data user.*

Thank you for the comment. The interpolation is certainly not a big deal for scientific users. However, since the dataset is also intended for other users (e.g. consulting offices, ministries), we decided to interpolate 6 hours linearly and export the interpolated timesteps in an additional text file "LamaH\D_gauges\2_timeseries\gaps". We kindly hope for your understanding.

**15.** *Fig 3. Figure 3a legend title should be 'Start of continuous data record'. I also think the x-axis on the histogram in Figure 3b is incorrect – should go from 0 – 100%?*

Caption of Figure 3a was changed, see our answer at 4). The fraction of gaps in each runoff time series (Figure 3b) are plotted in [‰]. In Fig. 3b the last class is set to >50‰ because only few gauges have more than 50‰ gap fraction in their time series.

**16.** *L190. I am unclear what you mean by 'gauge hierarchies'?*

The gauge hierarchy indicates the order of the gauge within the interconnected river network. Please find more information in the caption of Fig. 2) in 1.3). We hope that our response can clarify the ambiguity.

**17.** *L270. 'for each of the 3 different basin delineations' – you don't need 'different' here and can be worded as 'for each of the 3 basin delineations'. Also L423, L601 and L602 – you don't need the word 'different' here.*
Thank you for your comment, we removed the term "different" in many cases.

**18.** *L310. What do you mean by 'large notches in catchment shape'?*
We meant irregular incisions in the shape of the catchment area. We propose to remove this sentence, because it doesn't add relevant information for the users.

**19.** *L555. What do you mean by 'herding'?*
We propose to change the term to "weed".

**20.** *L605. I disagree that 'These uncertainties have been addressed'. They have been considered and discussed but I wouldn't say they have been addressed as many are not quantified in the dataset.*
Thank you very much, we have definitely not described all sources of uncertainty. We propose to change the sentence as follows:

It is clear that LamaH contains deficits and uncertainties due to the large number of data sources included. We however tried to consider and discuss most of these limitations.

---

## Author Response (AR2)

**Reply to the comments of the Editor on the manuscript "LamaH | Large-Sample Data for Hydrology and Environmental Sciences for Central Europe" submitted to ESSD (https://doi.org/10.5194/essd-2021-72).**

Dear Lukas Gudmundsson,

we have uploaded the revised LamaH dataset on Zenodo at the known repository (https://doi.org/10.5281/zenodo.4525244). Additionally, we have also corrected some small details in the manuscript. Differences between version4 and version3:

Line 7: "" replaced by "LamaH-CE"

Line 610: "" replaced by "ungauged"

Line 618: "" replaced by "gauged"

in Table C2: "" replaced by "Qobs"; "" replaced by "Gauged"

Yours sincerely,

Christoph in behalf of all authors.